



# Lessons from a high CO₂ world: an ocean view from ~3 million years ago

Erin L. McClymont[1], Heather L. Ford[2], Sze Ling Ho[3], Julia C. Tindall[4], Alan M. Haywood[4], Montserrat Alonso-Garcia[5,6], Ian Bailey[7], Melissa A. Berke[8], Kate Littler[7], Molly Patterson[9], Benjamin Petrick[10], Francien Peterse[11], A. Christina Ravelo[12], Bjørg Risebrobakken[13], Stijn De Schepper[13], George E.A. Swann[14], Kaustubh Thirumalai[15], Jessica E. Tierney[15], Carolien van der Weijst[11] and Sarah White[16].

[1]Department of Geography, Durham University, Durham, DH1 3LE, U.K.
[2]School of Geography, Queen Mary University of London, London, U.K.
[3]Institute of Oceanography, National Taiwan University, 10617 Taipei, Taiwan.
[4]School of Earth and Environment, University of Leeds, Leeds, LS29JT, U.K.
[5]Department of Geology, University of Salamanca, Salamanca, Spain.
[6]CCMAR, Universidade do Algarve, 8005-139 Faro, Portugal.
[7]Camborne School of Mines & Environment and Sustainability Institute, University of Exeter, Exeter, U.K.
[8]Department of Civil and Environmental Engineering and Earth Sciences, University of Notre Dame, Notre Dame IN 46656, USA.
[9]Department of Geological Sciences and Environmental Studies, Binghamton University SUNY, 4400 Vestal Pkwy E, Binghamton, New York USA.
[10]Max Planck Institute for Chemistry, Climate Geochemistry Department, 55128 Mainz, Germany.
[11]Department of Earth Sciences, Utrecht University, Utrecht, 3584 CB, the Netherlands.
[12]Department of Ocean Sciences, University of California, Santa Cruz, CA, USA,
[13]NORCE Norwegian Research Centre and Bjerknes Centre for Climate Research, 5007 Bergen, Norway.
[14]School of Geography, University of Nottingham, Nottingham, NG7 2RD, U.K.
[15]Department of Geosciences, The University of Arizona, Tucson, AZ 85721, USA
[16]Dept. of Earth and Planetary Sciences, University of California, Santa Cruz, USA.

*Correspondence to*: Erin L. McClymont (erin.mcclymont@durham.ac.uk), Heather L. Ford (h.ford@qmul.ac.uk), Sze Ling Ho (slingho@ntu.edu.tw).

**Abstract.** A range of future climate scenarios are projected for high atmospheric CO₂ concentrations, given uncertainties over future human actions as well as potential environmental and climatic feedbacks. The geological record offers an opportunity to understand climate system response to a range of forcings and feedbacks which operate over multiple temporal and spatial scales. Here, we examine a single interglacial during the late Pliocene (KM5c, ca. 3.205 +/- 0.01 Ma) when atmospheric CO₂ concentrations were higher than pre-industrial, but similar to today and to the lowest emission scenarios for this century. As orbital forcing and continental configurations were almost identical to today, we are able to focus on equilibrium climate system response to modern and near-future CO₂. Using proxy data from 32 sites, we demonstrate that global mean sea-surface temperatures were warmer than pre-industrial, by ~2.3 ℃ for the combined proxy data (foraminifera Mg/Ca and alkenones), or by ~3.2℃ (alkenones only). Compared to the pre-industrial, reduced meridional gradients and enhanced warming in the North Atlantic are consistently reconstructed. There is broad agreement between data and models at the global scale, with regional differences reflecting ocean circulation and/or proxy signals. An uneven distribution of proxy data in time and space





does, however, add uncertainty to our anomaly calculations. The reconstructed global mean sea-surface temperature anomaly
for KM5c is warmer than all but three of the PlioMIP2 model outputs, and the reconstructed North Atlantic data tend to align
with the warmest KM5c model values. Our results demonstrate that even under low $CO_2$ emission scenarios, surface ocean
warming may be expected to exceed model projections, and will be accentuated in the higher latitudes.

## 1 Introduction

By the end of this century, projected atmospheric $CO_2$ concentrations range from 430 to >1000 ppmv depending upon future
emission scenarios (IPCC, 2013). At the current rate of emissions, global mean temperatures are projected to exceed 1.5°C and
2°C above pre-industrial values in 10 and 20 years, respectively, passing the targets set by the Paris Agreement (IPCC, 2018).
The geological record affords an opportunity to explore key global and regional climate responses to different atmospheric
$CO_2$ concentrations, including those which extend beyond centennial timescales (Fischer et al., 2018). Palaeoclimate models
indicate that climates last experienced during the mid-Piacenzian stage of the Pliocene (3.1–3.3 Ma) will be surpassed by 2030
CE under high emission scenarios (Representative Concentration Pathway, RCP8.5), or will develop by 2040 CE and be
sustained thereafter under more moderate emissions (RCP4.5, Burke et al., 2018).

The late Pliocene thus provides a geological analogue for climate response to moderate $CO_2$ emissions. However, the
magnitude of tropical ocean warming differs between proxy reconstructions (e.g. Zhang et al., 2014; O'Brien et al., 2014; Ford
and Ravelo, 2019; Tierney et al., 2019a), and stronger polar amplification has been consistently recorded in proxy data
compared to models (Haywood et al., 2013; Haywood et al., 2016a). Some of the disagreements may reflect non-thermal
influences on temperature proxies (e.g. secular evolution of seawater Mg/Ca, Medina-Elizalde et al., 2008; Evans et al., 2016)
and/or seasonality in the recorded signals (e.g. Tierney and Tingley, 2018). It has also been proposed that previous approaches
to integrating Pliocene sea-surface temperature (SST) data may have introduced bias to data-model comparison (Haywood et
al., 2013). For example, the Pliocene Research Interpretation and Synoptic Mapping (PRISM) project generated warm peak
averages within specified time windows (Figure 1) (outlined in Dowsett et al., 2016 and references therein). However, by
integrating multiple warm peaks within the 3.1–3.3 Ma mid-Piacenzian data synthesis windows (Figure 1), regional and time-
transgressive responses to orbital forcing (Prescott et al., 2014; Fischer et al., 2018; Hoffman et al., 2017) are potentially
recorded in the proxy data, which may not align with the more narrowly-defined time interval being modelled (Haywood et
al., 2013; Dowsett et al., 2016).

Here, we present a new, globally-distributed synthesis of SST data for the mid-Piacenzian stage, addressing two concerns.
First, we minimise the impact of orbital forcing on regional and global climate signals by synthesising data from a specific
interglacial stage, a 20-kyr time slice centred on 3.205 Ma (KM5c; see Figure 1). At 3.205 Ma, both seasonal and regional
distributions of incoming insolation are close to modern, making this time an important analogue for 21$^{st}$ century climate



(Haywood et al., 2013). The low variability in orbital forcing through KM5c minimises the potential for time-transgressive regional signals to be a feature of the geological data (Haywood et al., 2013; Prescott et al., 2014). Second, we provide a range of estimates from different SST proxies, taking into consideration the uncertainties in proxy-to-temperature calibrations and/or secular processes that may bias proxy estimates. This synthesis is possible due to robust stratigraphic constraints placed on the
datasets by the PAGES-PlioVAR working group (see Methods).

## 2 Methods

### 2.1 The KM5c Interglacial

KM5c (also referred to as KM5.3) is an interglacial centred on a ~100 kyr window of relatively low benthic $\delta^{18}O$ values, that immediately follows a pronounced $\delta^{18}O$ peak during the glacial stage M2 (3.3 Ma; Figure 1). Minor changes to orbital forcing
during KM5 enables a wider target zone (3.205 Ma +/- 20 kyr) for data collection, because the potential for orbitally-forced regional and time-transgressive climate signals is minimised (Haywood et al., 2013). A comparable approach has been adopted by the PRISM4 synthesis (3.190 to 3.220 Ma, Foley and Dowsett, 2019, see Figure 1). Here, we focus on a narrow time slice of 3.195-3.215 Ma, to span approximately one precession cycle. The reconstructed atmospheric $CO_2$ concentrations from boron isotopes in KM5c are $360 \pm 55$ ppmv (for median boron-derived values (*n=3*), full range: 289-502 ppmv, Figure 1; Foster et
al., 2017). A wider range of atmospheric $CO_2$ concentrations has been reconstructed for the whole mid-Piacenzian stage (356 $\pm 65$ ppmv for median values (*n=36*), full range: 185-592 ppmv, Foster et al., 2017).

### 2.2 Age models

The PAGES-PlioVAR working group agreed on a set of stratigraphic protocols to maximise confidence in the identification and analysis of orbital-scale variability within the mid-Piacenzian stage (McClymont et al., 2017). Sites were only included in
the synthesis if they had either (i) ≤10 kyr resolution benthic $\delta^{18}O$ data which could be (or had been) tied to the LR04 stack (Lisiecki and Raymo, 2005) or the HMM-Stack (Ahn et al., 2017); and/or (ii) the palaeomagnetic tie-points for Mammoth top (C2An.2n (b) at 3.22 Ma) and Mammoth bottom (C2An.3n (t) at 3.33 Ma). At one site (ODP Site 1090) these conditions were not met (see Supplement), but tuning to LR04 had been made using a record of dust concentrations under the assumption that higher dust flux occurred during glacials as observed during the Pleistocene (Martinez-Garcia et al., 2011). At ODP Site 806,
uncertainty over age control resulted from the absence of an agreed splice across the multiple holes drilled by ODP, and a new age model has been constructed (see Supplement). For some sites (see online summary at https://pliovar.github.io/km5c.html), revisions to the published age model were made, for example if the original data had been published prior to the LR04 stack (Lisiecki and Raymo, 2005) or prior to revisions to the palaeomagnetic timescale (Gradstein et al., 2012). In total, data from 32 sites was compiled, extending from 46°S to 69°N (Figure 2).



### 2.3 Proxy sea-surface temperature (SST) data

A multi-proxy approach was taken, to maximise the information available on changing climates and environments during the KM5c interval. Two SST proxies were analysed: the alkenone-derived $U^K_{37}$' index (Müller et al., 1998) and foraminifera calcite Mg/Ca (Delaney et al., 1985). Both proxies have several calibrations to modern SST: here we explore the impact of calibration choice on KM5c SST data, by comparing and contrasting outputs between proxies and between calibrations. Although the $TEX_{86}$ proxy (Schouten et al., 2002) has also been used to generate mid-Piacenzian SSTs (e.g. O'Brien et al., 2014; Petrick et al., 2015; Rommerskirchen et al., 2011), this data is not included here because it could not be confidently assigned to the KM5c interval either due to low sampling resolution and/or our age control protocol was not met.

### 2.3.1 Alkenone SSTs (the $U^K_{37}$' index)

The majority of the 23 alkenone-derived sea-surface temperature (SST) datasets included in the PlioVAR synthesis used the $U^K_{37}$' index, and applied the linear core-top calibration (60°S-60°N) (Müller et al., 1998) (hereafter Müller98; Tables S2 and S3). The Müller98 calibration applies the best fit between core-top $U^K_{37}$' and modern SSTs, recorded at the sea surface (0 m water depth, 60°S to 60°N) and consistent with haptophyte productivity in the photic zone. The sedimentary signal is proposed to record annual mean SST based on linear regression (Müller et al., 1998). Cultures of one of the dominant haptophytes, *Emiliania huxleyi*, generated only minor differences in the slope of the $U^K_{37}$'-temperature relationship (Table S2), where growth temperature was used for calibration (Prahl et al., 1988). Several PlioVAR datasets were originally published using the Prahl et al. (1988) calibration (Table S3).

A recent expansion of the global core-top database (<70°N) was accompanied by Bayesian statistical analysis to assess the relationship(s) between predicted (from $U^K_{37}$') and recorded ocean temperatures (Tierney and Tingley, 2018). The revised $U^K_{37}$' calibration, BAYSPLINE, addresses non-linearity in the $U^K_{37}$'-SST relationship at the high end of the calibration, i.e. in the low-latitude oceans (Pelejero and Calvo, 2003; Sonzogni et al., 1997). BAYSPLINE also highlights scatter between predicted and observed SSTs at the high latitudes, and explicitly reconstructs seasonal SSTs >45°N (Pacific) and >48°N (Atlantic), and in the Mediterranean Sea (Tierney and Tingley, 2018).

To test the impact of different alkenone temperature calibrations on the quantification of mid-Piacenzian SSTs, we converted all $U^K_{37}$' data to SSTs using both the Müller98 calibration and the BAYSPLINE calibration. For most sites, BAYSPLINE was run with the recommended setting for the prior standard deviation scalar (pstd) of 10 (Tierney and Tingley, 2018). At high $U^K_{37}$' values (above ~24°C) it is recommended to use the more restrictive value of 5, to minimise the possibility of generating unrealistic SSTs (e.g. >40°C) given that the slope of the $U^K_{37}$'-temperature calibration becomes attenuated (Tierney and Tingley, 2018).



### 2.3.2 Foraminifera Mg/Ca

The magnesium-to-calcium ratio of foraminifera calcite can be used to reconstruct sea surface (surface dwelling), thermocline

(subsurface dwelling) and deep (benthic) ocean temperatures (Delaney et al., 1985;Elderfield et al., 1996;Rosenthal et al., 1997). The PlioVAR dataset includes analysis from 12 sites, on surface-dwelling foraminifera *Globigerinoides ruber*, *Trilobatus sacculifer*, and *Globigerina bulloides* (Table S4). In the original publications, data were converted to SST using a range of calibrations as well as corrections for $CaCO_3$ dissolution in the water column and sediments, which leads to preferential removal of Mg from the $CaCO_3$ lattice (generating cooler SSTs than expected; (Dekens et al., 2002; Regenberg et

al., 2006; Regenberg et al., 2009). Evolution of the Mg/Ca of seawater (Mg/Ca$_{seawater}$) over geological timescales may also impact Mg/Ca-based palaeo-temperature reconstructions (Brennan et al., 2013; Coggon et al., 2010; Fantle and DePaolo, 2005; Gothmann et al., 2015; Horita et al., 2002; Lowenstein et al., 2001). Changes in Mg/Ca$_{seawater}$ impacts the intercept and potentially the sensitivity of palaeotemperature equations (Evans and Müller, 2012; Medina-Elizalde and Lea, 2010), but there remains uncertainty over the magnitude of Mg/Ca$_{seawater}$ changes in the late Pliocene (O'Brien et al., 2014; Evans et al., 2016).


To test the impact of different foraminifera Mg/Ca SST calibrations on mid-Piacenzian SSTs, we compare published SSTs with the recently developed BAYMAG calibration (Tierney et al., 2019b). We use published SSTs because the original researchers used their best judgement to choose a particular Mg/Ca-SST calibration, given that it (i) fitted modern (regional) core-top values; (ii) accounted for known environmental impacts (e.g. $[CO_3^{2-}]$ correction); (iii) was developed within a

particular research group; and/or (iv) fitted conventional wisdom at the time. BAYMAG uses a Bayesian approach that incorporates laboratory culture and core-top information to generate probabilistic estimates of past temperatures. BAYMAG assumes a sensitivity of Mg/Ca to salinity, pH and saturation site at each core site, and also accounts for Mg/Ca$_{seawater}$ evolution through a linear scaling (i.e. there is no change in the sensitivity of the palaeo-temperature equation as Mg/Ca$_{seawater}$ evolves) (Tierney et al., 2019b). For each site with Mg/Ca data, we computed SSTs using BAYMAG's species-specific hierarchical

model. In the absence of knowledge concerning changes in salinity, pH, and saturation state in the Pliocene, we assumed that these values were the same as today. We drew seasonal sea-surface salinity from the World Ocean Atlas 2013 product (Boyer et al., 2013), and pH and bottom water saturation state from the GLODAPv2 product (Lauvset et al., 2016; Olsen et al., 2016). We used a prior standard deviation of 6˚C for all sites.

**2.4 Climate models**

The model outputs used here were generated from the 15 models that contribute to the Pliocene modelling intercomparison project, Phase 2 (PlioMIP2) (Haywood et al., submitted). The boundary conditions for the experiments, and their large-scale results for Pliocene and pre-industrial climates are detailed elsewhere (Haywood et al., submitted; Haywood et al., 2016b), so are briefly outlined here.






The Pliocene simulations are intended to represent KM5c (~3.205Ma) and were forced with PRISM4 boundary conditions (Haywood et al., 2016b). Atmospheric $CO_2$ concentration was set at 400 ppmv (Haywood et al., submitted), in line with the upper estimates of atmospheric $CO_2$ from boron isotope data (Figure 1; Foster et al., 2017). Lower estimates from the alkenone carbon isotope proxy (Figure 1) are likely to reflect an insensitivity of this proxy to atmospheric $CO_2$ in the Pliocene (Badger

et al., 2019). All other trace gases, orbital parameters and the solar constant were specified to be consistent with each model's preindustrial experiment. The Greenland Ice Sheet was confined to high elevations in the Eastern Greenland Mountains, covering an area approximately 25% of the present-day ice sheet. The Antarctic ice sheet has no ice over Western Antarctica. The reconstructed PRISM4 ice sheets have a total volume of $20.1 \times 10^6$ km$^3$, equating to a sea-level increase relative to present day of less than ~24 m (Dowsett et al., 2016).


Modelling groups had some choices regarding exact implementation of boundary conditions, however 14 of the 15 models used the 'enhanced' PRISM4 boundary conditions (Dowsett et al., 2016) which included all reconstructed changes to the land/sea mask and ocean bathymetry. Initialisation of the experiments varied between models (Haywood et al., submitted). Some models were initialised from a pre-industrial state while others were initialised from the end of a previous Pliocene

simulation or another warm state. The integration length of the simulations was between 500 and 4000 years.

### 2.5 Statistical analysis (calculating of global means / gradients)

For all anomaly calculations we obtain pre-industrial SST from the NOAA-ERSST5 dataset for years 1870–1899 CE (Huang et al., 2017), ensuring alignment between the KM5c proxy data and the KM5c model experiments (Haywood et al., submitted).

The global mean SST anomaly from the proxy data was obtained as follows: firstly, the SST anomaly between the proxy data and the NOAA-ERSST5 data was obtained for each location, and the data collated into bins of 15° of latitude. It is assumed that the average of all the data in each bin represents the average SST anomaly for that latitude band. Next, the area of the ocean surface for each bin is obtained. The average SST anomaly is then the average of all the bins weighted by the ocean area in the relevant latitude band.


Meridional gradients were obtained in a similar way. A low-latitude SST anomaly was obtained as the weighted average of all the bins containing low-latitude SSTs (for example the $4 \times 15°$ bins containing latitudes of 30°S - 30°N), and a high-latitude SST anomaly was obtained as the weighted average of all bins containing high latitude SSTs (>60°N, because there were no proxy data points >60°S). The meridional gradient SST anomaly is then the low-latitude SST anomaly minus the high-

latitude SST anomaly, relative to the pre-industrial.

There are some errors in this calculation of the global mean SST, in particular, the fact that the proxy data is not evenly distributed throughout a latitude bin, and also that some bins contain very few data points. There is a higher density of data in



the Atlantic Ocean, compared to the Indian Ocean and Pacific Ocean, and no high-latitude data is available to consider a

Southern Ocean response (Figure 2). Nevertheless, this method of calculating averages does attempt to account for unevenly distributed data and provides a SST anomaly (SSTA) that is comparable with model results. The impact of proxy choice was examined in the calculation of the global means and meridional SST gradients. As no Mg/Ca data were available >50°N or >30°S, we calculated global mean SST and the meridional SST gradients either including or excluding the Mg/Ca data; both results are outlined below and shown in Table 1.

## 3 Results

Relative to the pre-industrial, the combined $U^{K}_{37}$' and Mg/Ca proxy data, using the original calibrations, indicate a KM5c global mean SST anomaly of +2.3°C and a meridional SST gradient reduced by 2.6°C (Figure 3). The amplitude of the global SST mean anomaly in the combined proxy data exceeds those indicated in all but three of the PlioMIP2 models, whereas the meridional temperature gradient anomalies are comparable (Figure 3). Overall, the proxy data show the lowest temperature

anomalies in the low latitudes, regardless of proxy (from +3°C to -4°C for sites <30°N/S). A larger range of temperature anomalies is reconstructed in the mid- and high-latitudes (from +9°C to -2°C for sites >30°N/S) (Figure 4). Thus, there is a broad, but complex, pattern of enhanced warming at the mid- and high-latitudes, reflecting a combination of regional influences on circulation patterns, and to some extent, proxy choice. This pattern is not explained by temporal variability nor sample density within the KM5c time interval: regardless of sample number per site, the standard deviation is <1.5°C (Figure S4). We

note that of the 32 sites examined here, 7 provided a single data point for the KM5c interval (Figure S2, alkenones: ODP Sites 907, 1081, U1337, U1417; Figure S3, foraminifera Mg/Ca: DSDP Sites 214, 709, 763); the sites are geographically well distributed, however, and so unlikely to significantly impact our global mean / gradient calculations.

Calibration choice has a small impact over the reconstructed patterns of KM5c SST anomalies (Figure 4, and Figures S2 and

S3). Below 24°C, absolute $U^{K}_{37}$' SSTs using Müller98 are <1°C lower than those using BAYSPLINE. At high temperatures the non-linearity in the BAYSPLINE calibration means that BAYSPLINE-SSTs can be up to 1.67 °C ± 0.01°C higher than when using Müller98 (Figure S2). The low latitude offset between Müller98 and BAYSPLINE has two effects: it elevates the global mean SST (Figure 3, Table 1) and increases the KM5c meridional SST gradient towards pre-industrial values (Figures 3 and 4, Table 1). The calibration offsets are less systematic for Mg/Ca. There is a wider range of offsets between BAYMAG

and published SST values (from -4 to +5°C, Figure S3 Table S3), although the smallest KM5c SST anomalies continue to be reconstructed in the low-latitudes, regardless of which Mg/Ca calibration is applied (Figure 4).

Overall, the $U^{K}_{37}$'-temperature anomalies lie within the range given by PlioMIP2 models (Figure 4). The Mg/Ca estimates are mainly from the low latitudes, and high-latitude (>60°N/S) Mg/Ca SST data are not available to calculate meridional gradients

using foraminifera data alone (Figure 4). Mg/Ca-SST anomalies are generally lower than for $U^{K}_{37}$', and a cooler KM5c than



pre-industrial is consistently (but not always) recorded in the low-latitudes by Mg/Ca regardless of calibration choice (Figure 4). As a result, combining $U^{K}_{37}$' and Mg/Ca data leads to a cooler global mean SST (~2.3°C) than when using $U^{K}_{37}$' alone (~3.2°C, Figure 3). At 8 sites, the negative KM5c SST anomalies in Mg/Ca disagree with both the $U^{K}_{37}$' data and the PlioMIP2 model outputs (Figure 4). Only three sites have both $U^{K}_{37}$' and Mg/Ca data (DSDP Site 609, IODP Sites U1313 and U1143);

reconstructed SSTs for IODP Sites U1313 and U1143 are within calibration uncertainty while DSDP Site 609 has colder Mg/Ca estimates than alkenones or models (Figure 4).

## 4 Discussion

### 4.1 SST expression of the KM5c interglacial

KM5c is characterised by a surface ocean which is ~2.3°C (alkenones and Mg/Ca) or ~3.2°C (alkenones-only) warmer than
pre-industrial, with a ~2.6°C reduction in the meridional SST gradient. The global mean SST anomaly is higher than the 1.7°C previously calculated for the wider mid-Piacenzian warm period (3.1-3.3 Ma), regardless of proxy choice (IPCC, 2014b). Previous analysis of a suite of models suggested that a climate state resembling the mid-Piacenzian was likely to develop and be sustained under RCP4.5 (Burke et al., 2018). The PlioMIP2 multi-model mean indicates that mid-Piacenzian warming is more comparable to projections for RCP6.0 by 2100 CE (Haywood et al., submitted). Our proxy-based mean global SST
anomaly is larger than most PlioMIP2 models (Figure 3); because air temperature increases are larger over land than over the ocean in PlioMIP2 models (Haywood et al., submitted), our results suggest that the global annual surface air temperature anomaly for KM5c likely exceeds the PlioMIP2 multi-model surface air temperature mean of 2.8°C. The higher global SST mean recorded in the KM5c proxy data, compared to the PlioMIP2 models, occurs despite the available atmospheric $CO_2$ reconstructions indicating values below the ~400 ppmv used in the PlioMIP2 models (Figure 1). Our synthesis of SST data
thus indicates that with atmospheric $CO_2$ concentrations ≤400 ppmv (comparable to RCP4.5), the surface ocean warming response will likely be larger than indicated in models. Further work is required to increase the temporal resolution of the atmospheric $CO_2$ reconstructions through KM5c, to improve our understanding of the reconstructed SST response to $CO_2$ forcing, including whether (or by how much) the reconstructed atmospheric $CO_2$ differ from model boundary conditions.

Proxy choice, calibration choice, and site selection, all have an impact on the magnitude of the change in meridional SST gradient for KM5c compared to the pre-industrial (Table 1). Focussing only on a Northern Hemisphere SST gradient leads to higher gradient anomalies than when all of the low-latitudes are included (30°S-30°N), because it excludes the high SST anomalies of the Benguela upwelling sites (20-25°S, discussed below, Figure 4). Smaller meridional SST gradient anomalies occur using BAYSPLINE (+0.03 to -1.66°C) than the Muller98 calibration for $U^{K}_{37}$' (-1.18 to -3.00°C, Table 1), due to the
increased low-latitude SST anomalies generated by BAYSPLINE (Figure 4). Due to several (but not all) low-latitude sites recording negative SST anomalies for KM5c using foraminifera Mg/Ca, the inclusion of Mg/Ca data leads to a larger difference





in the meridional SST gradient relative to the pre-industrial (-2.19 to -4.08°C). Further work is required to fully understand the negative KM5c SST anomalies in some of the low-latitude sites (discussed further below), given their impact on the meridional SST gradients. However, a robust pattern emerging from the data is that the KM5c proxy data detail smaller low-latitude SST

anomalies than those of the mid- and high-latitudes SST anomalies (Figure 4), leading to a reduction in the meridional SST gradient relative to the pre-industrial. Enhanced mid- and high-latitude warming has been observed in other warm intervals of the geological past, including the last interglacial and the Eocene (Evans et al., 2018; Fischer et al., 2018), and is a feature of future climate under elevated $CO_2$ concentrations(IPCC, 2014a).

There is complexity in the amplitude of the KM5c SST anomaly by latitude and basin, which may reflect patterns of surface ocean circulation. In the Northern Hemisphere, relatively muted warming in the East Greenland Current (ODP Site 907, 69°N) may reflect the presence of at least seasonal sea-ice cover from c.4.5 Ma (Clotten et al., 2018). In contrast, relatively high SST anomalies at ODP Sites 642 (67°N) and 982 (58°N) track northward flow of the North Atlantic Current, accounting for the enhanced warming relative to North-east Pacific IODP Site U1417 (57°N; Figures 4 and 5). The large North Atlantic SST

anomalies also contribute to an enhanced Northern Hemisphere meridional SST gradient of up to 4°C (>60°N minus 0–30°N; Table 1). For the Southern Hemisphere, a signal of polar amplification is less clearly identified than for the Northern Hemisphere (Figure 4), although we recognise that all sites are <46°S. Low KM5c SST anomalies (<2°C) at DSDP Sites 593 and 594 (41 °S and 46 °S, respectively) might be accounted for by a similar positioning of the Subtropical Front close to New Zealand during KM5c as today (McClymont et al., 2016;Caballero-Gill et al., 2019). Antarctic Intermediate Water (AAIW)

temperatures were also only ~2.5°C warmer than pre-industrial during KM5c, suggesting a small warming in subantarctic waters where AAIW forms (McClymont et al., 2016). In contrast, large anomalies at ODP Sites 1125 (43 °S, Pacific) and 1090 (43°S, Atlantic) reflect a greater sensitivity to expanded subtropical gyres during KM5c, contrasting with the Pleistocene equatorward displacement (and enhanced cooling) of subpolar water masses (e.g. Martinez-Garcia et al., 2010) which today places both of these sites poleward of the Subtropical Front.


Given that our proxy data meridional SST gradient calculations use only two sites to calculate the high-latitude SSTs (ODP Sites 907 and 642), which are also both from the Nordic Seas (Figure 2), we explored the impact of expanding our high-latitude band into the mid-latitudes. We also explored narrowing the low-latitude band so that it does not include the Benguela upwelling sites, which have a significant data-model offset (Figure 4) and may be influenced by localised circulation changes

(see section 4.2). Previous calculations of Pliocene meridional SST gradients have also considered differences between the mid and low latitudes through time (Fedorov et al., 2015). Despite adding 4 more sites by expanding the high-latitude band to 45°N/S, the meridional SST gradients are reduced by <0.4°C, from -1.18 to -1.56°C using the original $U^K_{37}$' data (Table 1). However, it is clear from the distribution of sites (Figure 2) that our reconstructed KM5c SSTs (and thus the global mean and meridional gradients) have a strong signal from the Atlantic Ocean. There is a relative scarcity of sites from the Indian Ocean,

Pacific Ocean and Southern Ocean, but it is difficult to ascertain what impact this may have had on our global analysis. Further



work is required to increase the spatial density of SST data for KM5c and the wider mid-Piacenzian stage, to better evaluate the magnitude of the warming and gradient changes outlined here.

**4.2 Proxy data-model comparisons for mid- and high-latitude sites**

For the mid- and high-latitudes, we find broad proxy data/model agreements for most sites. In the North Atlantic Ocean, reconstructed SST KM5c anomalies from $U^K_{37}$' fall within the ranges provided by the PlioMIP2 models (Figure 4) for all but one site (IODP Site U1387, 37°N). The overall $U^K_{37}$'-model agreement for the North Atlantic Ocean suggests that, as proposed by Haywood et al. (2013), a focus on a specific interglacial within the mid-Piacenzian provides an improved comparison to the climate being simulated by the PlioMIP2 models. Thus, some of the data-model mismatch in previous mid-Piacenzian

syntheses (e.g. Dowsett et al., 2012) may have been due to the averaging of warm peaks which may not have been synchronous in time between sites and/or with the interval being modelled. Disagreements occur between proxies (North Atlantic) and between proxies and models (Benguela upwelling, Gulf of Cadiz / Mediterranean Sea) (Figure 4). Here, we explore the potential causes for these offsets in turn.

The largely $U^K_{37}$'-derived data from the North Atlantic tend to align with the warmest model outputs (Figure 4), and the $U^K_{37}$'-SST anomalies also tend to be larger than those from Mg/Ca. A challenge for understanding the cause(s) of the $U^K_{37}$'-Mg/Ca differences is that only two sites have data from both proxies, and these do not show a consistent signal. There is good correspondence between $U^K_{37}$' and the original published Mg/Ca SSTs for IODP Site U1313 (41°N), whereas at ODP Site 609 Mg/Ca SSTs (both calibrations) are between 4.6-6.1°C cooler than $U^K_{37}$'. It has also been shown that the $U^K_{37}$'-Mg/Ca SST

offset at Site 609 is not constant with time for the late Pliocene (Lawrence and Woodard, 2017). When using BAYMAG, warmer KM5c SSTs are reconstructed than the original published data at DSDP Site 603 and IODP Site U1313 (35°N and 41°N, Figure 4), but BAYMAG reconstructs SSTs only 0.8°C warmer than from the original published SSTs at Site 609. These mid-latitude North Atlantic Mg/Ca data are provided by *G. bulloides*, which may calcify at depth in the water column (e.g. Mortyn and Charles, 2003; Schiebel et al., 1997), and account for those sites where Mg/Ca reconstructions give lower

reconstructed SSTs than from $U^K_{37}$' (Bolton et al., 2018; De Schepper et al., 2013). Alternatively, an offset between alkenones and Mg/Ca might be accounted for if there is a seasonal bias to the $U^K_{37}$' calibration (e.g. Conte et al., 2006; Schneider et al., 2010). Despite documented seasonality in alkenone production at high latitudes, it has been proposed that mean annual SSTs continue to be recorded by $U^K_{37}$' in sediments (Rosell-Melé and Prahl, 2013), as indicated by the original $U^K_{37}$' calibration (Müller et al., 1998). In contrast, BAYSPLINE explicitly assumes an autumn signal is recorded in Atlantic sites >45ºN (Tierney

and Tingley, 2018). Despite these differences in interpretation, BAYSPLINE values for KM5c are <0.7ºC cooler than the original published $U^K_{37}$' data (Figure S2). Although all of the North Atlantic $U^K_{37}$' data align with the mean annual SST anomalies generated by the PlioMIP2 models (Figure 4), three of the sites show alignment between $U^K_{37}$'-SSTs and the July-





November values from the multi-model means (Figure 5). In contrast, Site 907 aligns with cool spring temperatures in the models, perhaps reflecting production after sea ice melt.


The large data-model discrepancy at 30°S reflects 3 sites which today sit beneath the Benguela upwelling system in the South-East Atlantic (20-26°S, Figure 4). Part of the data-model discrepancy in the KM5c anomaly can be attributed to the models over-estimating pre-industrial SSTs in the northern Benguela sites (NOAA-ERSST5 SSTs are 2–5°C below the pre-industrial model range), and suggests that models are not fully capturing the local dynamics of the coastal upwelling today (Small et al.,

2015). Realistic representations of the Benguela upwelling system today are proposed to require realistic wind stress curl and high-resolution atmosphere and ocean models (<1°, Small et al., 2015). Most of the PlioMIP2 simulations use lower resolution atmosphere and ocean models (Haywood et al., submitted). An increased density of proxy data reconstructing KM5c atmospheric circulation, as well as application of high-resolution models, may help to understand the observed KM5c data-model discrepancy. Furthermore, there was a deep thermocline during the Pliocene (as reconstructed in the Equatorial Pacific;

(Ford and Ravelo, 2019; Ford et al., 2015; Steph et al., 2006; Steph et al., 2010); and theorized globally (Philander and Fedorov, 2003)), so that warmer subsurface waters than today were upwelled, enhancing local warming. However, warming of ~3.4°C in subsurface waters (Ford and Ravelo, 2019) and ~2.5°C in intermediate waters (McClymont et al., 2016) for Pliocene interglacials suggest that Pliocene upwelling of warmer waters is unable to fully account for the 7–10°C SST anomalies in Benguela sites for KM5c. Changes to the distribution of export productivity and SSTs indicate that an overall poleward

displacement of the Benguela Upwelling system occurred during the Pliocene, so that the main zone of upwelling likely sat close to ODP Site 1087 at 31°S (Etourneau et al., 2009; Petrick et al., 2018; Rosell-Melé et al., 2014). As the northern and southern Benguela regions are today marked by differences in the seasonality of the upwelling, a temporal shift in upwelling intensity may also account for some of the large SST anomaly (Haywood et al., submitted). Thus, the data-model disagreement may be accounted for by a combination of displaced upwelling and warmer upwelled waters, giving large SST anomalies in

Benguela proxy data, alongside the challenges of modelling both the pre-industrial and KM5c upwelling system and its associated SSTs.

Data-model disagreement also occurs at two northern hemisphere sites where $U^{K}_{37}$'-SST anomalies exceed model predictions (Figure 4). Punto Piccola (Sicily, 37°N) is located within the Mediterranean Sea, whereas IODP Site U1387 (37°N) records

the influence of the southward flow of the subtropical gyre in the Gulf of Cadiz. The data-model disagreement for KM5c reflects warmer SST estimates from the proxy data compared to the models, despite the good agreement for the pre-industrial suggesting that locally complex ocean circulation in these near-shore and marginal marine settings may have been captured in the models. The data-model offset is also likely to be a minimum, because BAYSPLINE Mediterranean SSTs explicitly record November-May temperatures (Tierney and Tingley, 2018), and alkenone production below the sea surface has also been

proposed (Ternois et al., 1997): both scenarios would act to raise mean annual SSTs further from those simulated in the PlioMIP2 models (Figure 4). Further multi-proxy investigation is required to identify whether the data-model disagreements





in Benguela upwelling, Gulf of Cadiz, and Mediterranean Sea reflect challenges in modelling near-shore or complex oceanographic systems and/or biases in the temperature signal recorded by the proxy data.

### 4.3 Data-model comparisons for low-latitude sites


The low-latitude $U^{K}_{37}$'-SST anomalies for KM5c align well with the PlioMIP2 models (Figure 4). At ODP Sites 806 and 959, Mg/Ca anomalies range between +0.2 to +1.3°C compared to the pre-industrial (Figure 4) and also align with the PlioMIP2 models. Only one low-latitude site has both $U^{K}_{37}$' and Mg/Ca SST data: ODP Site 1143 (9°N) records KM5c anomalies of +0.8 to +2.5°C ($U^{K}_{37}$') or -4 to -9 °C (Mg/Ca). Although the $U^{K}_{37}$' data align with the model outputs for Site 1143, the negative

anomaly in Mg/Ca lies outside the model range for mean annual SST (Figure 4).

Six of the low-latitude sites have negative low-latitude SST anomalies in KM5c from foraminifera Mg/Ca; these occur regardless of whether the original or BAYMAG calibrations are applied, and for both *G. ruber* and *T. sacculifer*-based reconstructions. The negative KM5c Mg/Ca-SST anomalies lie beyond those shown across the PlioMIP2 model range (Figure

4), despite the absolute Mg/Ca-SSTs reconstructed from these sites for KM5c falling within the model range for all but two of the sites (ODP Sites 999 (13°N) and 1241 (6°N); Figure S5). However, the absolute SST values reconstructed for KM5c from Mg/Ca tend to align with the colder model outputs (Figure S5).

Mg/Ca-SST calibration choice has no consistent impact on the KM5c anomalies (across all latitudes, Figure 4). Therefore, the

corrections for secular seawater Mg/Ca change and/or non-thermal influences over Mg/Ca, which are accounted for in BAYMAG (Tierney et al., 2019b) do not account for these cold tropical KM5c anomalies. For example, for ODP Site 806 in the Western Pacific warm pool, BAYMAG SST estimates for KM5c are ~1°C warmer than the published Mg/Ca record (Wara et al., 2005). For Site 999 in the Caribbean Sea, BAYMAG SST estimates for KM5c are ~0.5°C cooler than the published Mg/Ca record (De Schepper et al., 2013). This also suggests the impact of Mg/Ca$_{seawater}$ change on SST is small on warm pool

sites. The Mg/Ca$_{seawater}$ correction used in BAYMAG is conservative, drawing on multiple lines of physical evidence (corals, fluid inclusions, calcite veins, etc) (Tierney et al., 2019b). Given the variable directions of the offsets between published and BAYMAG SSTs shown here, the Mg/Ca$_{seawater}$ correction is unable to account for the data-model offsets observed for the low latitudes.

$CaCO_3$ dissolution in the water column and sediments could lead to a cool bias on the Mg/Ca-SSTs (Dekens et al., 2002; Regenberg et al., 2006; Regenberg et al., 2009). However, the cool KM5c anomalies also occur if the forward-modelled core-top Mg/Ca SSTs from BAYMAG are used as the pre-industrial 'reference' (Figure S6). The cold low-latitude anomalies for KM5c could reflect an increase in the calcification depth of the foraminifera, since the surface-dwelling foraminifera analysed here calcify at a range of depths, particularly in the tropics where the thermocline is deep in comparison to mid- to high-





latitudes (Fairbanks et al., 1982;Curry et al., 1983). The negative anomalies are broadly smaller for *G. ruber* (-0.4 to -1.2°C) than for *T. sacculifer* (-0.6 to -3.5°C), consistent with a deeper depth-habitat for the latter (Curry et al., 1983). There is a lack of consistency between sites, however, which is difficult to resolve when single species have been analysed for each of the sites through KM5c.

Where there are very large differences between BAYMAG and published Mg/Ca SST estimates, regardless of latitude (e.g. North Atlantic, Figure 4), we suggest that some combination of calibration difference, Mg/Ca$_{seawater}$ change and/or other environmental factors including seasonality and calcification depth may offer an explanation. To fully investigate the cause(s) of offsets in Mg/Ca SST reconstructions requires future multi-species analysis for Mg/Ca for each site, and multi-proxy analysis for each site. Such an approach would enable exploration of a wider range of potential influences over both the Mg/Ca

and U$^{K}_{37}$' SST reconstructions, and a reduction in the uncertainties of the reconstructed SSTs and their anomalies. Alongside foraminifera Mg/Ca and U$^{K}_{37}$' analyses, additional proxies which are likely to add valuable information about water column structure and seasonality could include TEX$_{86}$ (Schouten et al., 2002), long-chain diols (Rampen et al., 2012), and clumped isotopes (Zaarur et al., 2013). Resolving the causes of the different proxy-proxy and proxy-model offsets is important, because it impacts the calculation of the global mean SST anomaly relative to pre-industrial; however, even with inclusion of the overall

cooler Mg/Ca data, the combined KM5c proxy data still indicate a global mean SST anomaly which is larger than most models from the PlioMIP2 experiments (Figure 3).

## 5 Conclusions

This study has generated a new multi-proxy synthesis of SST data for an interglacial stage (KM5c) from the Pliocene. By

selecting an individual interglacial, with orbital forcing similar to modern, we are able to focus on the SST response to atmospheric CO$_2$ concentrations comparable to today and the near-future (~400 ppmv), but elevated relative to the pre-industrial. Using strict stratigraphic protocols we selected only those data which could be confidently aligned to KM5c. By comparing different calibrations and two different proxy systems (U$^{K}_{37}$' and Mg/Ca in planktonic foraminifera) we identified several robust signals which are proxy-independent. First, global mean SSTs during KM5c were warmer than pre-industrial.

Second, there was a reduced meridional SST gradient which is the result of relatively small low-latitude SST anomalies and a larger range of warming anomalies for the mid- and high-latitudes. Overall, there is good data-model agreement for both the absolute SSTs and the anomalies relative to the pre-industrial, although there are complexities in the results. Further work is required to generate multi-proxy SST data from single sites, accompanied by robust reconstructions of thermocline temperatures using multi-species foraminifera analysis, so that the range of factors explaining proxy- and calibration-offsets

can be explored more fully.

The choice of proxy for SST reconstruction impacts the overall calculation of global mean SST and the meridional gradients. The negative anomalies in Mg/Ca-SSTs in six of the sixteen low-latitude sites lowers the global mean SST of KM5c from





~3.2ºC ($U^K_{37}$'-only) to ~2.3ºC (combined $U^K_{37}$' and Mg/Ca). The meridional SST gradient anomalies are decreased to -2.6ºC

(combined $U^K_{37}$' and Mg/Ca) relative to the pre-industrial, although a more muted reduction (up to -1.18°C) occurs with $U^K_{37}$' alone. A number of factors may lead to a cool bias in the foraminifera Mg/Ca SSTs, which require further investigation through multi-proxy and multi-species analysis, particularly in low-latitude sites.

We identify the strongest warming across the North Atlantic region. The results are consistent with the PlioMIP2 models,

although the largely $U^K_{37}$' data sit at the high end of the calculated model anomalies. Although seasonality may play a role in the proxy data signal, these results also suggest that many models may under-estimate high-latitude warming even with the moderate $CO_2$ increases identified in KM5c relative to the pre-industrial. More data points are required to fully explore these patterns: for seven sites only one data point lay within KM5c, and more than half of the analysed sites (18/32) recorded Atlantic Ocean SSTs.


Both the PlioMIP2 models (Haywood et al., submitted) and future projections (IPCC, 2018) indicate that warming is higher over land than in the oceans in response to higher atmospheric $CO_2$ concentrations. Our synthesis of KM5c thus likely represents a minimum warming to be expected with atmospheric $CO_2$ concentrations of ~400 ppmv. Even under low $CO_2$ emission scenarios, our results demonstrate that surface ocean warming may be expected to exceed model projections, and

will be accentuated in the higher latitudes.

## 6 Data availability

Most of the original datasets used here can be downloaded from the NOAA (www.ncdc.noaa.gov) and PANGAEA (www.pangaea.de) data repositories. Full details of data sources can be accessed at https://pliovar.github.io/km5c.html. The

combined proxy data (absolute SST reconstructions and anomalies to Pre-industrial) will be available at www.pangaea.de (*awaiting confirmation of the data URL, December 2019*).

## 7 Supplement link

Additional information on proxy calibrations and their impact on the SST reconstructions can be accessed at (*link from CPD*).


## 8 Author contribution

ELM, HLF and SLH designed the data analysis and led the data compilation. JCT and AMH processed outputs from the suite of PlioMIP2 models, and calculated global means and meridional SST gradients using the proxy data. Proxy data were compiled and their age models reviewed by ELM, HLF, MA-G, IB, KL, MP, BP, ACR, BR, SDS, GEAS, KT, and SW. Proxy

calibrations were reviewed and applied by ELM, MAB, HLF, SLH, FP, JET, and CvdW. ELM, HLF and SLH prepared the manuscript with contributions from all co-authors.



## 9 Competing interests

The authors declare they have no conflict of interest.


## 10 Acknowledgements

This work is an outcome from the several workshops sponsored by Past Global Change (PAGES) as contributions to the working group on Pliocene Climate Variability over glacial-interglacial timescales (PlioVAR). We acknowledge PAGES for their support and the workshop participants for discussions. Funding support has also been provided by NERC (NE/I027703/1

and NE/L002426/1 to ELM), Leverhulme Trust (Philip Leverhulme Prize, ELM), and the Research Council of Norway (BR and ELM (221712), SDS (229819)). FP and CvdW are part of the Netherlands Earth System Science Centre (NESSC), financially supported by the Dutch Ministry of Education, Culture and Science (OCW). MAG acknowledges support from FCT (SFRH/BPD/96960/2013, PTDC/MAR-PRO/3396/2014 and CCMAR UID/Multi/04326/2019). This research used samples and/or data provided by the International Ocean Discovery Program (IODP), Ocean Drilling Program (ODP) and

Deep Sea Drilling Project (DSDP).

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





**Figure 1: the KM5c interglacial during the late Pliocene (3.195-3.215 Ma). Upper panel: benthic oxygen isotope stack (solid line: LR04 (Lisiecki and Raymo, 2005); dashed line and grey shading: Prob-stack mean and 95% confidence interval, respectively, (Ahn et al., 2017)). Selected Marine Isotope Stages (KM2 through M2) are highlighted. The KM5c interval of focus here is indicated by the shaded blue bar. Previous Pliocene synthesis intervals are also shown: PRISM3 (3.025–3.264 Ma) and PRISM4 (isotope stages KM5c-M2; (Dowsett et al., 2016)); (c) reconstructed atmospheric CO₂ concentrations (Foster et al., 2017). Points show mean reported data (except white crosses: median values from Martinez-Boti et al., 2015), shading shows reported upper and lower estimates. Past and projected atmospheric CO₂ concentrations highlighted by arrows: PlioMIP2 simulations are run with CO₂ at 400 ppmv (Haywood et al., submitted) close to the annual mean in 2018 (NOAA), Pre-Industrial values from ice cores (Loulerge et al., 2008) and projected representative concentration pathways (RCP) for 2100 CE (IPCC, 2013).**




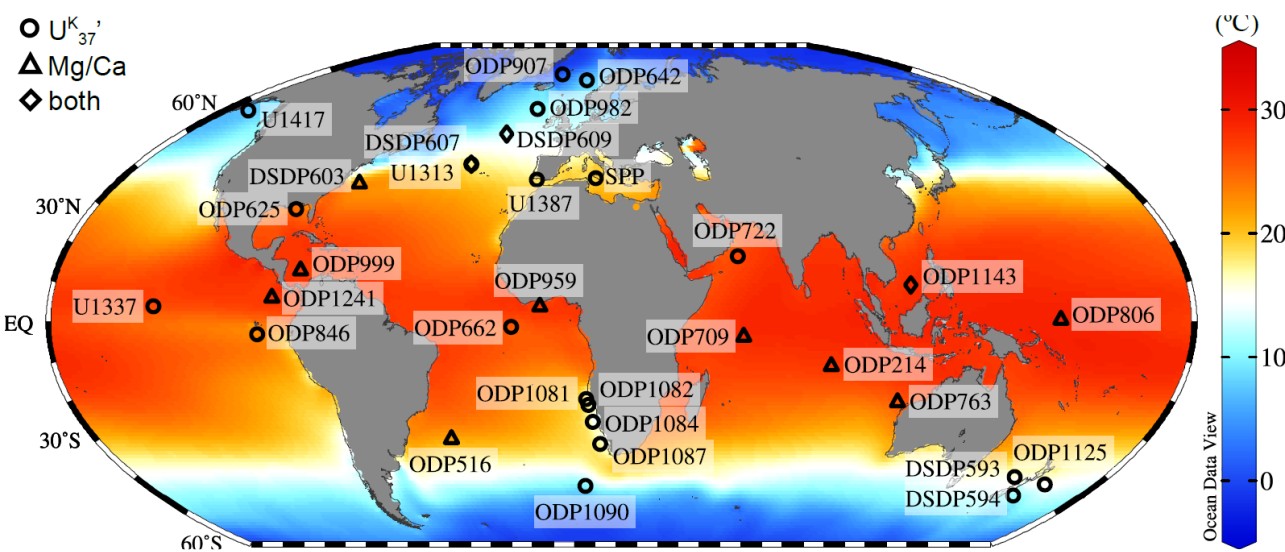

**Figure 2: Locations of sites used in the synthesis. A full list of the data sources and proxies applied per site can be accessed at**

https://pliovar.github.io/km5c.html**.**





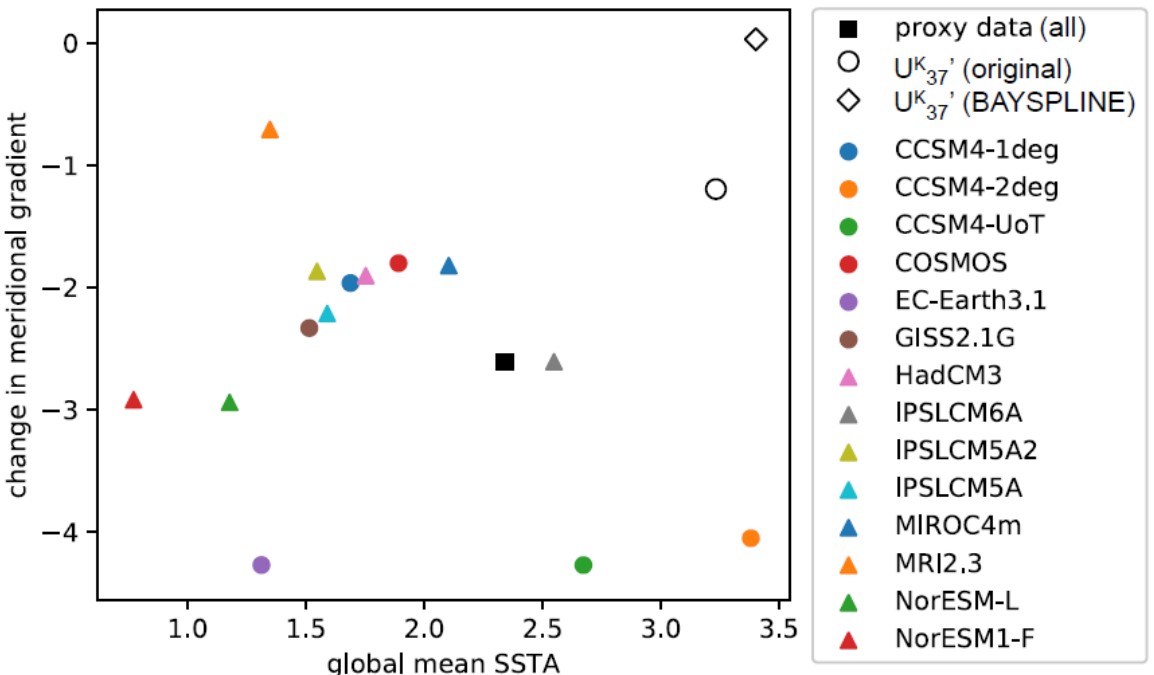

**Figure 3: Comparison of KM5c SST data relative to pre-industrial (NOAA-ERSST5) for global mean SST anomalies (SSTA) and the change in meridional SST gradient, constructed using proxy data and the suite of PlioMIP2 models. The meridional SST gradient is calculated as 30°S-30°N minus 60°N-75°N, so that a more negative change in the gradient reflects a larger warming anomaly at high latitudes relative to low latitudes. Proxy data calculations were made using either all proxy data (UK37' and Mg/Ca using their original calibrations), or using only U$^K_{37}$' data. No Mg/Ca data is available >60°N so we were unable to calculate Mg/Ca-only gradients (Figure 3). The impact of changing the low- and high-latitude bands is explored in Table 1.**





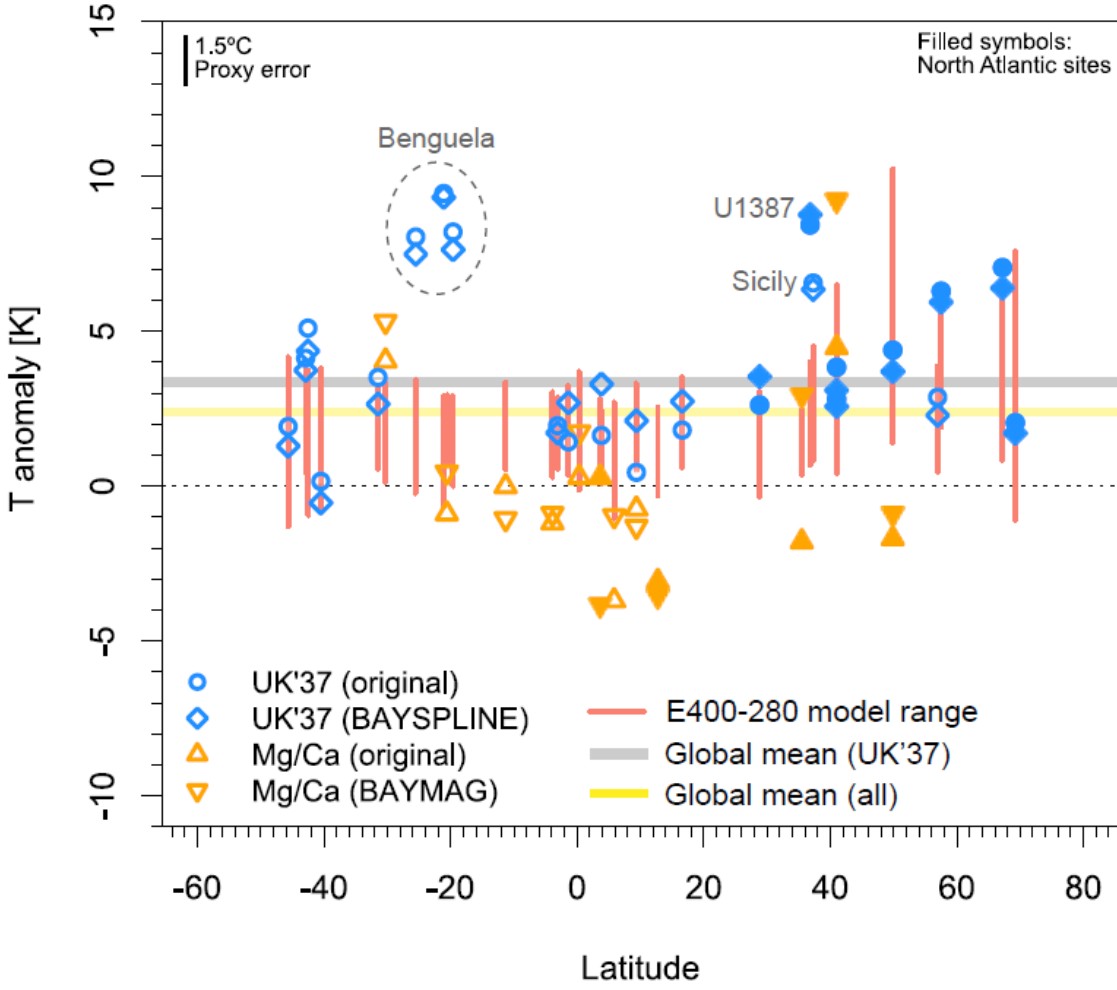

**Figure 4: reconstructed and modelled SST anomalies plotted by latitude. SST reconstructions using the original published data and two Bayesian approaches (BAYSPLINE, BAYMAG) are shown. Vertical red lines show the range of modelled annual SSTs from all PlioMIP2 experiments (Haywood et al., submitted) calculated at the grid boxes containing each site.**






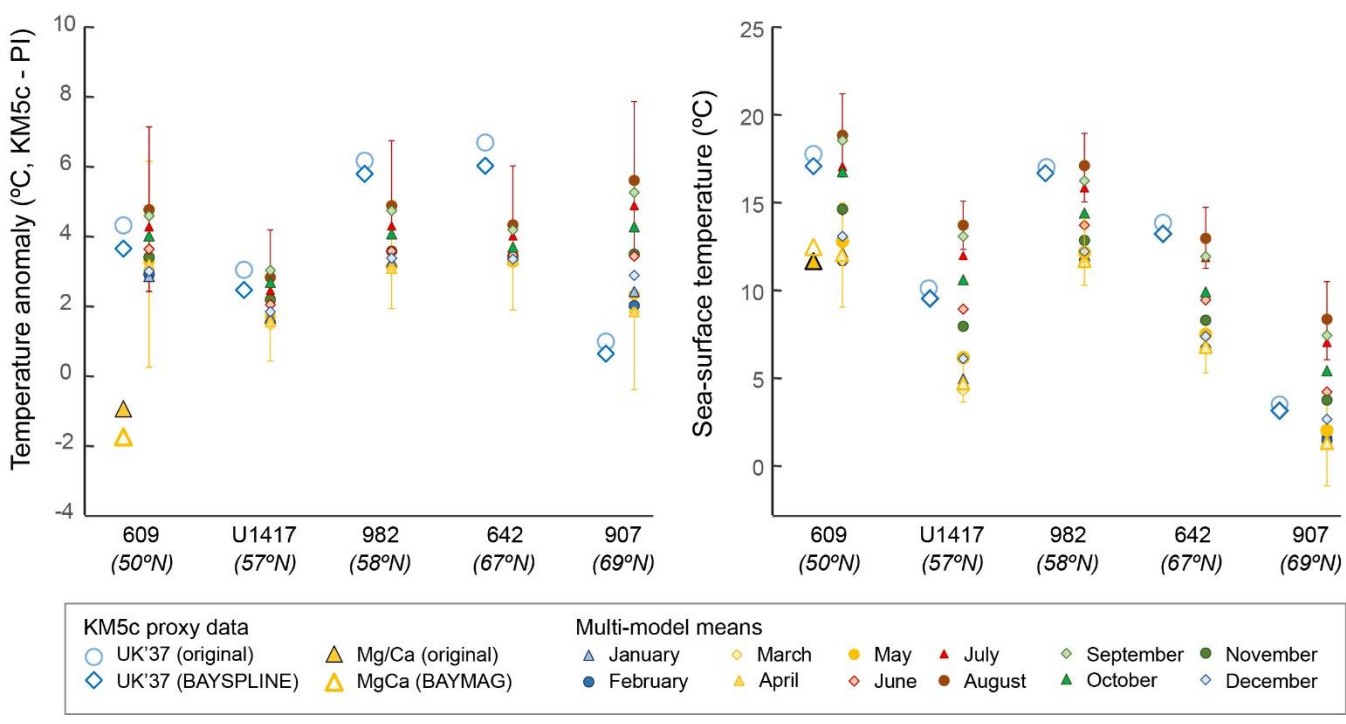

**Figure 5. Investigating the potential seasonal signature recorded in high latitude Northern Hemisphere sites (>50ºN), ordered by increasing latitude from left to right. Note that Site U1417 is from the North Pacific, where BAYSPLINE explicitly assumes a summer signal is recorded >48ºN. All other sites are from the Atlantic Ocean / Nordic Seas, where BAYSPLINE assumes an autumn signal >45ºN. The original calibration by Müller et al. (1998) proposes that mean annual SSTs are recorded. Standard deviations of the multi-model means are shown for August (red) and April (yellow), which tend to be the maxima and minima, respectively.**







**Table 1: Comparison of the magnitude of the global SST anomaly and meridional SST gradients between KM5c and pre-industrial, depending on proxy combination, and the latitudinal bands used for the gradient calculations.**

| Proxy | Global mean SST anomaly, °C | Meridional SST gradient anomaly, °C | | |
|---|---|---|---|---|
| | | 30°S-30°N minus >60°N | 0-30°N minus >60°N | 15°S-15°N minus >45°N/S |
| U$^K_{37}$' (original) | 3.24 | -1.18 | -3.00 | -1.56 |
| U$^K_{37}$' (BAYSPLINE) | 3.41 | 0.03 | -1.66 | -0.18 |
| U$^K_{37}$' (original) + Mg/Ca | 2.32 | -2.61 | -4.08 | -2.84 |
| U$^K_{37}$' (BAYSPLINE) + Mg/Ca (BAYMAG) | 2.28 | -2.21 | -3.13 | -2.19 |
