# Peer review of "Lessons from a high CO2 world: an ocean view from ~3 million years ago"

_Climate of the Past, 2019_

## Referee Comment (RC1) · Antje Voelker (Referee) · 3 Mar 2020

McClymont and co-authors compiled a data base of sea-surface temperature (SST) data for the KM5c interglacial period during the late Pliocene and use that data to evaluate temperature changes in the Pliocene relative to the pre-industrial period and to climate modeling results for the late Pliocene warm period. The authors made a thorough (re-)evaluation of the age models of the sediment records included in their compilation and also converted the Uk37' values to SST applying the same calibrations (Mueller et al., 1998 and the newer BAYSPLINE). The Mg/Ca trace element based temperature reconstructions are also presented with the SST values reconstructed originally and with BAYMAG. The reconstructed SST values are a thoroughly evaluated including pointing out potential bias in the proxy data or caused by local hydrographic

conditions. The manuscript is well-written, suitable for publication in Climate of the Past and presents a major step-forward in data-model comparison for the late Pliocene warm period.

I only have a few minor comments for the authors that could help improve the manuscript.

1) The references for the original data for each Site are provided on the Pliovar webpage, which is a very informative tool. However, from past experience with "local" data bases, I am wondering for how long this link will be maintained, i.e. can the authors guarantee that this link still exists in 5 or 10 years. So I would like to see a "paper" version as Table S1, especially since the age model related information is not necessarily "hidden" in the references listed in Tables S3 and S4.

2) p. 5 line 152: correct site to state after saturation

3) p. 6 line 183: I suggest to include a short comment clarifying that the pre-industrial period selected has no overlap with the Little Ice Age.

4) p. 11 line 355: since there is evidence for the existence of Mediterranean Outflow Water (MOW) during the Pliocene along the southern Iberian margin, one should expect the Azores Current, whose existence is linked to the formation of MOW in the Gulf of Cadiz (see for example Oezgoekmen, T.M., Chassignet, E.P., Rooth, C.G.H., 2001. On the connection between the Mediterranean Outflow and the Azores Current. Journal of Physical Oceanography 31, 461-480), also to be present. Nowadays, the subtropical surface waters in the Gulf of Cadiz seem to be more derived from the Azores Current and its northern branches extending into the Gulf of Cadiz and towards the SW Iberian margin than the southward gyre recirculation (Portugal Current). So besides the southward recirculation you would also have the direct across-North Atlantic basin transport between 32 and 36°N and those waters might be warmer than the southward recirculation. Most models do not resolve the MOW, so the (heat) transport associated with the Azores Current might also not exist. I also suggest to include

(Iberian margin) behind Gulf of Cadiz because not every reader will know where the Gulf of Cadiz is located.

5) Fig. S5: mention in the figure caption what the gray envelope represents.

6) Fig. S2 and S3: with the start of IODP (2003-2013) program Site names include a letter to identify the platform with which they were drilled. So correctly, it should say U1313, U1337, U1387 and U1417.

---

## Referee Comment (RC2) · Tim Herbert (Referee) · 10 Mar 2020

This paper presents a very significant compilation of SST for a time window in the late Pliocene based on two of the most validated SST proxies. In comparison to previous reconstructions for the "PRISM interval", it benefits from greatly improved stratigraphic control, thus minimizing spurious variance or patterns from chronological dispersion. A major concern I had as a reviewer is that, given the numerous data-model comparisons of the PRISM/PlioMIP consortia, what would be distinctive? This is answered successfully on page 10: "The overall UK 37'-model agreement for the North Atlantic Ocean suggests that, as proposed by Haywood et al. (2013), a focus on a specific interglacial within the mid- Piacenzian provides an improved comparison to the climate being simulated by the PlioMIP2 models. Thus, some of the data-model mismatch in

previous mid-Piacenzian syntheses (e.g. Dowsett et al., 2012) may have been due to the averaging of warm peaks which may not have been synchronous in time between sites and/or with the interval being modelled" The prime contention is that the Pliocene data imply that: "Even under low CO2 emission scenarios, our results demonstrate that surface ocean warming may be expected to exceed model projections, and will be accentuated in the higher latitudes. " I think the paleo-sensitivity field is likely to oscillate for some time between the view that paleo-SST/CO2 constraints confirm the Charney sensitivity of 2.4-3.5oC for a CO2 doubling versus the view (implied here) that the deep time/Earth System sensitivity is considerably higher. Jessica Tierney argues that the current models are right based on her LGM reconstructions; I'm not so sure. This paper certainly favors a higher Earth System sensitivity, without an explicit numerical figure. I find myself a bit embarrassed that I do not have more critical comments to the manuscript. It is very clearly organized and written, and anticipates most objections along the way. Ambiguities are clearly recognized, for example in paleo-CO2 estimates. Here the authors handle nuances such as likely insensitivity of alkenone CO2 proxy with good judgement. The paper also includes multiple assessments of uncertainties, including geographic bias, the effects of including/excluding Mg/Ca. Examples include: increasing # of sites in the high latitude band to check on the implied reduction of meridional SST gradients, and recognizing the data bias to the Atlantic. The major uncertainty seems to be whether to use alkenone BAYSPLINE, which enhances the global warming and increases the meridional SST gradient Specific text comments: I suggest changing "a ∼100 kyr window of relatively low benthic  18O values" to "a ∼100 kyr window of relatively depleted benthic $\delta$18O values" to avoid the possible ambiguity of what "low" means to the reader. P 7: "Thus, there is a broad, but complex, pattern of enhanced warming at the mid- and high-latitudes, reflecting a combination of regional influences on circulation patterns, and to some extent, proxy choice. This pattern is not explained by temporal variability nor sample density within the KM5c time interval: regardless of sample number per site, the standard deviation is <1.5oC (Figure S4). " In light of the "complex pattern" and "temporal variability", can

the authors clarify what the standard deviation" refers to? I assume it's the variation at a given site within the KM5c time bin? So I suggest adding "the standard deviation at any site within the time bin is < 1.5oC" P. 7: "Mg/Ca-SST anomalies are generally lower than for UK 37', ". Since the authors have been pointing out differences in the interpretation of Uk'37 anomalies between using the Muller linear regression and the BAYSPLINE calibration, can they clarify whether the Mg/Ca anomalies are lower than BOTH calibrations or specifically the BAYSPLINE one or both? From context, given that 8 sites gave a negative KM5c anomaly (!!) It would seem that Mg/Ca deviates from both alkenone calibrations. p. 8: "KM5c is characterised by a surface ocean which is ∼2.3°C (alkenones and Mg/ Ca) or ∼3.2°C (alkenones-only) warmer than 240 pre-industrial, with a ∼2.6°C reduction in the meridional SST gradient. " Which alkenone calibration was adopted for the "alkenones-only" estimate? What would be the difference of Muller vs BAYSPLINE? p. 11: I think there's something funny about the NOAA-ERSST temperatures for the region. We have unpublished alkenone SST estimates from Site 1085 for the KM5c interval that show an anomaly of ∼3 degrees when using the WOAA. My sense is that the large Benguela anomalies arise entirely from using the NOAA-ERSST atlas and that they would fall in line with expected values if other atlases were used. The authors should at a minimum consult other atlases and explore the possibility of a regional SST bias in the NOAA-ERSST estimates. I think this is a much more parsimonious explanation than the oceanographic ones proposed in lines 340-352. This in fact is my major suggestion: to examine whether that data base imposes a significant bias to the results here. In summary, I recommend publication with minor revisions (excepting my last point). This is a very nice piece of work.

---

## Author Comment (AC1) · 30 Apr 2020

We thank the reviewer for the positive and constructive comments on the manuscript. Here we reply to the reviewers comments:

Reviewer comment 1) The references for the original data for each Site are provided on the Pliovar webpage, which is a very informative tool. However, from past experience with "local" data bases, I am wondering for how long this link will be maintained, i.e. can the authors guarantee that this link still exists in 5 or 10 years. So I would like to see a "paper" version as Table S1, especially since the age model related information is not necessarily "hidden" in the references listed in Tables S3 and S4.

REPLY: The reviewer raises a valid concern. After the submission of our data

to PANGAEA, and whilst this review was in progress, we were subsequently required to provide full references for all data sources. As a result, the information stored on our PlioVAR webpage will also be available via PANGAEA if this manuscript is accepted for publication. Our revised PANGAEA link is: https://doi.pangaea.de/10.1594/PANGAEA.911847 This has now been corrected in the placeholder statement in the submitted manuscript (line 482).

Reviewer comment 2) p. 5 line 152: correct site to state after saturation

REPLY: This has been corrected.

Reviewer comment 3) p. 6 line 183: I suggest to include a short comment clarifying that the pre-industrial period selected has no overlap with the Little Ice Age.

REPLY: There are several definitions in the literature for the timing of the Little Ice Age. Our selection of the years 1870-1899 CE as pre-industrial overlaps with the final decades of the most broadly defined LIA (1440-1920 CE; Owens et al., 2017), but after the time of the greatest cooling and before the onset of 20th century warming (Owens et al., 2017; PAGES2k Consortium, 2017). We have added a line to clarify this in the main text (lines 205-206).

Reviewer comment 4) p. 11 line 355: since there is evidence for the existence of Mediterranean Outflow Water (MOW) during the Pliocene along the southern Iberian margin, one should expect the Azores Current, whose existence is linked to the formation of MOW in the Gulf of Cadiz (see for example Oezgoekmen, T.M., Chassignet, E.P., Rooth, C.G.H., 2001. On the connection between the Mediterranean Outflow and the Azores Current. Journal of Physical Oceanography 31, 461-480), also to be present. Nowadays, the subtropical surface waters in the Gulf of Cadiz seem to be more derived from the Azores Current and its northern branches extending into the Gulf of Cadiz and towards the SW Iberian margin than the southward gyre recirculation (Portugal Current). So besides the southward recirculation you would also have the direct across-North Atlantic basin transport between 32 and 36N and those waters

might be warmer than the southward recirculation. Most models do not resolve the MOW, so the (heat) transport associated with the Azores Current might also not exist. I also suggest to include (Iberian margin) behind Gulf of Cadiz because not every reader will know where the Gulf of Cadiz is located.

REPLY: Thanks to the reviewer for highlighting the link between the Azores Current, Mediterranean Outflow Water, and potential influences on the sea-surface temperatures at site U1387. We agree that the position and temperature of the Azores Current could be an important contributor to differences between model outputs and data, and future work could investigate similarities and differences between models for these two systems. We have edited the text to flag the potential influence of the cross-basin transport by the Azores Current, noting also here that our original statement about the local complexity of ocean circulation in this area still stands (lines 384-386). We have clarified in the text that site U1387 is in the "Gulf of Cadiz, Iberian margin" (lines 385-385).

Reviewer comment 5) Fig. S5: mention in the figure caption what the gray envelope represents.

REPLY: This has been corrected (the gray envelope represents the range of sea-surface temperatures recorded at each latitude). This figure is S6 in the revised manuscript.

Reviewer comment 6) Fig. S2 and S3: with the start of IODP (2003-2013) program Site names include a letter to identify the platform with which they were drilled. So correctly, it should say U1313, U1337, U1387 and U1417.

REPLY: This has been corrected.

Literature cited: Owens, M.J. et al. (2017) The Maunder minimum and the Little Ice Age: an update from recent reconstructions and climate simulations. Journal of Space Weather and Space Climate, 7, article A33, doi: 10.1051/swsc/2017034. PAGES2k

Consortium (2017) A global multiproxy database for temperature reconstructions of the Common Era. Scientific Data 4:170088 doi: 10.1038/sdata.2017.88.

---

## Author Response (AR1)

**CP-2019-161** *Lessons from a high CO2 world: an ocean view from ~3 million years ago*

**Reply to Editor**

This document combines the published responses we made to our two reviewers, and the additional author comment we made available. We also include the "tracked changes" manuscript text and the "tracked changes" supplementary text.

We thank the Editor for the remaining comment regarding the comparison of different datasets for calculating pre-industrial anomalies. We had showed an additional comparison in our response to reviewer 2: we agree with the Editor that this would be well placed within the manuscript, and have included as an expanded Supplementary Figure 7.

In addition to addressing the reviewer's comments we identified some minor errors during the editing process which have also been corrected (e.g. the numbering of graphics in the Supplemental). These edits are all visible in the "tracked changes" document.

**CP-2019-161 *Lessons from a high CO2 world: an ocean view from ~3 million years ago***

**Author comment**

We thank both reviewers for their positive reviews and constructive comments. We have replied to each of their concerns in the accompanying replies.

Here we wish to provide two further updates to the manuscript, linked to the ongoing modelling efforts as part of the PlioMIP2 initiative.

(1) Since the submission of our manuscript, an error in the calculation of the sea surface temperature anomaly (SSTA) in the NorESM model was identified. This has been corrected in the revised manuscript. The EC-Earth3.1 model was also withdrawn, and the results of the EC-Earth3.3 and CESM2 models have been added. These changes to the modelling outputs have been incorporated into our figures and statistical analysis, but have not required changes to the manuscript text. The following figures have been adjusted from our original submission: Figures 3, 4, 5, S5, S6 and S7.

(2) In our original submission we also anticipated that the model outputs we were presenting had already been available elsewhere (Haywood et al., in review, 2020). However, this was not the case for all of the model data which we presented. As a result, we have included as new co-authors those who were responsible for the design and implementation of the PlioMIP2 model experiments, and who processed the outputs which were then used for both the statistical analysis and the data-model comparisons. The author list has therefore been extended to include those significant contributions (see below). All co-authors have contributed to the writing and editing of the manuscript.

Literature cited:

Haywood, A. M., Tindall, J. C., Dowsett, H. J., Dolan, A. M., Foley, K. M., Hunter, S. J., Hill, D. J., Chan, W.-L., Abe-Ouchi, A., Stepanek, C., Lohmann, G., Chandan, D., Peltier, W. R., Tan, N., Contoux, C., Ramstein, G., Li, X., Zhang, Z., Guo, C., Nisancioglu, K. H., Zhang, Q., Li, Q., Kamae, Y., Chandler, M. A., Sohl, L. E., Otto-Bliesner, B. L., Feng, R., Brady, E. C., von der Heydt, A. S., Baatsen, M. L. J., and Lunt, D. J.: A return to large-scale features of Pliocene climate: the Pliocene Model Intercomparison Project Phase 2, Clim. Past Discuss., https://doi.org/10.5194/cp-2019-145, in review, 2020.

Additional co-authors and their affiliations (ordered according to the manuscript formatting):

Ayako Abe-Ouchi[17,18], Michiel L.J. Baatsen[19], Esther Brady[20], Wing-Le Chan[17], Deepak Chandan[21], Ran Feng[22], Chuncheng Guo[13], Anna S. von der Heydt[19], Stephen Hunter[4], Xiangyi Li[13,23], Gerrit Lohmann[24], Kerim H. Nisancioglu[13,25,26], Bette L.Otto-Bliesner[20], W. Richard Peltier[21], Christian Stepanek[24] and Zhongshi Zhang[13,27,28].

4. School of Earth and Environment, University of Leeds, Leeds, LS29JT, U.K.

13. NORCE Norwegian Research Centre and Bjerknes Centre for Climate Research, 5007 Bergen, Norway.

17. Atmosphere and Ocean Research Institute, The University of Tokyo, Kashiwa, 277-8564, Japan.

18. National Institute for Polar Research, Tachikawa, 190-8518, Japan.

19. Institute for Marine and Atmospheric research Utrecht (IMAU), Department of Physics, Utrecht University, Utrecht, 3584CC, The Netherlands.

20. Climate and Global Dynamics Laboratory, National Center for Atmospheric Research (NCAR), Boulder, 80305, USA.

21. Department of Physics, University of Toronto, Toronto, M5S1A7, Canada.

22. Department of Geosciences, University of Connecticut, Storrs, 06033, USA.

23. Climate Change Research Center, Institute of Atmospheric Physics, Chinese Academy of Sciences, Beijing 100029, China.

24. Alfred-Wegener-Institut - Helmholtz-Zentrum für Polar and Meeresforschung, Bremerhaven, 27570, Germany.

25. Department of Earth Science, University of Bergen, Allégaten 70, 5007 Bergen, Norway.

26. Centre for Earth Evolution and Dynamics, University of Oslo, Po. Box 1028 Blindern, 0315 Oslo, Norway.

27. Department of Atmospheric Science, School of Environmental Studies, China University of Geosciences, Wuhan, China.

28. Nansen-Zhu International Research Centre, Institute of Atmospheric Physics, Chinese Academy of Sciences, Beijing 100029, China.

**CP-2019-161** *Lessons from a high CO2 world: an ocean view from ~3 million years ago*

**Reply to reviewer 1**

We thank the reviewer for the positive and constructive comments on the manuscript. In response to the minor comments raised by reviewer 1 (which are italicised):

1) *The references for the original data for each Site are provided on the Pliovar webpage, which is a very informative tool. However, from past experience with "local" data bases, I am wondering for how long this link will be maintained, i.e. can the authors guarantee that this link still exists in 5 or 10 years. So I would like to see a "paper" version as Table S1, especially since the age model related information is not necessarily "hidden" in the references listed in Tables S3 and S4.*

The reviewer raises a valid concern. After the submission of our data to PANGAEA, and whilst this review was in progress, we were subsequently required to provide full references for all data sources. As a result, the information stored on our PlioVAR webpage will also be available via PANGAEA if this manuscript is accepted for publication.

Our revised PANGAEA link is: https://doi.pangaea.de/10.1594/PANGAEA.911847

This has now been corrected in the placeholder statement in the submitted manuscript (line 482).

2) *p. 5 line 152: correct site to state after saturation*
This has been corrected.

3) *p. 6 line 183: I suggest to include a short comment clarifying that the pre-industrial period selected has no overlap with the Little Ice Age.*
There are several definitions in the literature for the timing of the Little Ice Age. Our selection of the years 1870-1899 CE as pre-industrial overlaps with the final decades of the most broadly defined LIA (1440-1920 CE; Owens et al., 2017), but after the time of the greatest cooling and before the onset of 20th century warming (Owens et al., 2017; PAGES2k Consortium, 2017). We have added a line to clarify this in the main text (lines 205-206).

4) *p. 11 line 355: since there is evidence for the existence of Mediterranean Outflow Water (MOW) during the Pliocene along the southern Iberian margin, one should expect the Azores Current, whose existence is linked to the formation of MOW in the Gulf of Cadiz (see for example Oezgoekmen, T.M., Chassignet, E.P., Rooth, C.G.H., 2001. On the connection between the Mediterranean Outflow and the Azores Current. Journal of Physical Oceanography 31, 461-480), also to be present. Nowadays, the subtropical surface waters in the Gulf of Cadiz seem to be more derived from the Azores Current and its northern branches extending into the Gulf of Cadiz and towards the SW Iberian margin than the southward gyre recirculation (Portugal Current). So besides the southward recirculation you would also have the direct across-North Atlantic basin transport between 32 and 36N and those waters might be warmer than the southward recirculation. Most models do not resolve the MOW, so the (heat) transport*

*associated with the Azores Current might also not exist. I also suggest to include (Iberian margin) behind Gulf of Cadiz because not every reader will know where the Gulf of Cadiz is located.*

Thanks to the reviewer for highlighting the link between the Azores Current, Mediterranean Outflow Water, and potential influences on the sea-surface temperatures at site U1387. We agree that the position and temperature of the Azores Current could be an important contributor to differences between model outputs and data, and future work could investigate similarities and differences between models for these two systems. We have edited the text to flag the potential influence of the cross-basin transport by the Azores Current, noting also here that our original statement about the local complexity of ocean circulation in this area still stands (lines 384-386).

We have clarified in the text that site U1387 is in the "Gulf of Cadiz, Iberian margin" (lines 385-385).

*5) Fig. S5: mention in the figure caption what the gray envelope represents.*
This has been corrected (the gray envelope represents the range of sea-surface temperatures recorded at each latitude). This figure is S6 in the revised manuscript.

*6) Fig. S2 and S3: with the start of IODP (2003-2013) program Site names include a letter to identify the platform with which they were drilled. So correctly, it should say U1313, U1337, U1387 and U1417.*
This has been corrected.

Literature cited:

Owens, M.J. et al. (2017) The Maunder minimum and the Little Ice Age: an update from recent reconstructions and climate simulations. Journal of Space Weather and Space Climate, 7, article A33, doi: 10.1051/swsc/2017034.

PAGES2k Consortium (2017) A global multiproxy database for temperature reconstructions of the Common Era. Scientific Data 4:170088 doi: 10.1038/sdata.2017.88.

**CP-2019-161 *Lessons from a high CO2 world: an ocean view from ~3 million years ago***

**Reply to reviewer 2**

We thank the reviewer for the positive and constructive comments on the manuscript. In response to the minor comments raised by reviewer 2 (which are italicised):

1) *I suggest changing "a ~100 kyr window of relatively low benthic $\delta$18O values" to "a ~100 kyr window of relatively depleted benthic $\delta$18O values" to avoid the possible ambiguity of what "low" means to the reader.*

This has been corrected (line 98). We have edited to ensure that the statement is clear about which isotope is depleted:

"...centred on a ~100 kyr window of relatively depleted benthic $^{18}$O values..."

2) *P 7: "Thus, there is a broad, but complex, pattern of enhanced warming at the mid- and high-latitudes, reflecting a combination of regional influences on circulation patterns, and to some extent, proxy choice. This pattern is not explained by temporal variability nor sample density within the KM5c time interval: regardless of sample number per site, the standard deviation is <1.5oC (Figure S4). " In light of the "complex pattern" and "temporal variability", can the authors clarify what the standard deviation" refers to? I assume it's the variation at a given site within the KM5c time bin? So I suggest adding "the standard deviation at any site within the time bin is < 1.5oC"*

The reviewer is correct in his interpretation of the standard deviation plot. We have incorporated the reviewer's suggested change to the text (lines 235-236).

3) *P. 7: "Mg/Ca-SST anomalies are generally lower than for UK 37', ". Since the authors have been pointing out differences in the interpretation of Uk'37 anomalies between using the Muller linear regression and the BAYSPLINE calibration, can they clarify whether the Mg/Ca anomalies are lower than BOTH calibrations or specifically the BAYSPLINE one or both? From context, given that 8 sites gave a negative KM5c anomaly (!!) It would seem that Mg/Ca deviates from both alkenone calibrations.*

There is an inconsistent relationship between the alkenone SST anomalies and those from Mg/Ca. As we state, in general the Mg/Ca anomalies are lower than for UK'37, but at two sites we find Mg/Ca-SST anomalies which are greater than those of the model outputs (U1313 in the North Atlantic, Site 763 in the Indian Ocean). For both of these sites the original Mg/Ca calibration is lying closest to the Muller98 calibration output for sites in a similar latitude: but for Site 763 we note here that the 'closest' site by latitude is Site 1087 in the South Atlantic. This demonstrates the challenge we face for making direct comparisons between the proxy outputs: there are only three sites with both Mg/Ca and alkenone data available (which we stated in the same paragraph).

We recognise the reviewers concern that it would be useful to state this complexity more clearly. We have edited the text accordingly (lines 250-265):

Overall, the $U^{K}_{37}$'-temperature anomalies lie within the range given by PlioMIP2 models (Figure 4). The Mg/Ca estimates are mainly from the low latitudes, and high-latitude (>60ºN/S) Mg/Ca SST data are not available to calculate meridional gradients using foraminifera data alone (Figure 4). Mg/Ca-SST anomalies are generally lower than for $U^{K}_{37}$', and a cooler KM5c than pre-industrial is consistently (but not always) recorded in the low-latitudes by Mg/Ca regardless of calibration choice (Figure 4). As a result, combining $U^{K}_{37}$' and Mg/Ca data leads to a cooler global mean SST (~2.3°C) than when using $U^{K}_{37}$' alone (~3.2°C, Figure 3). At 8 sites, the negative KM5c SST anomalies in Mg/Ca disagree with both the $U^{K}_{37}$' data and the PlioMIP2 model outputs (Figure 4). The disagreement is present regardless of whether the Müller98 or BAYSPLINE calibrations are applied, but the difference is larger in the low latitudes for BAYSPLINE because this calibration generates higher SST values here (Section 2.3.1). Only three sites have both $U^{K}_{37}$' and Mg/Ca data (DSDP Site 609, IODP Sites U1313 and U1143) to enable direct comparison between Mg/Ca and alkenone SST data. Reconstructed SSTs for IODP Sites U1313 and U1143 are within calibration uncertainty. At Site U1313 (41°N) there is overlap between both alkenone outputs (Müller98 21.6°C, BAYSPLINE 20.9°C) and the original Mg/Ca reconstruction (22.2°C), whereas BAYMAG generates warmer SSTs (27.0°C). At Site 1143 (9°N), BAYSPLINE-SSTs are warmer (30.6°C) than from the Müller98 (28.9°C), original-Mg/Ca (27.7°C) and BAYMAG (27.1°C). In contrast, DSDP Site 609 (49°N) has colder Mg/Ca estimates (original 11.7°C, BAYMAG 12.5°C) than alkenones (Müller98 17.7°C, BAYSPLINE 17.1°C) or models (Figure 4).

4) p. 8: "KM5c is characterised by a surface ocean which is ~2.3 C (alkenones and Mg/ Ca) or ~3.2 C (alkenones-only) warmer than 240 pre-industrial, with a ~2.6 C reduction in the meridional SST gradient. " Which alkenone calibration was adopted for the "alkenones-only" estimate? What would be the difference of Muller vs BAYSPLINE?

The alkenones-only global SST anomaly we stated was generated using only the Müller calibration. For clarity, we now indicate in the text (lines 288-289) the two values:

"...~3.2°C (alkenones-only, Müller 98) or ~3.4°C (alkenones-only, BAYSPLINE calibration)...".

The abstract has also been edited to reflect this (line 56):

"...or by ~3.2-3.4°C (alkenones only)."

5) p. 11: I think there's something funny about the NOAA-ERSST temperatures for the region. We have unpublished alkenone SST estimates from Site 1085 for the KM5c interval that show an anomaly of ~3 degrees when using the WOAA. My sense is that the large Benguela anomalies arise entirely from using the NOAA-ERSST atlas and that they would fall in line with expected values if other atlases were used. The authors should at a minimum consult other atlases and explore the possibility of a regional SST bias in the NOAA-ERSST estimates. I think this is a much more parsimonious explanation than the oceanographic ones proposed in lines 340-352. This in fact is my major suggestion: to examine whether that data base imposes a significant bias to the results here.

The reviewer raises a concern about the alkenone SST anomalies we show for the Benguela upwelling region, which far exceed the anomalies calculated from the models (Figure 4 in the main text, and below). We have compared the data anomalies generated using NOAA-ERSST or the World Ocean Atlas 2018 (Locarnini et al., 2018), and show the results. Using WOA18 reduces the two of the SST anomalies in the Benguela upwelling sites: Site 1082 (from +9.5°C to +8.0°C) and Site 1081 (from +8°C to +6.5°C). In contrast, at Site 1084 there is an increase in the SST anomaly by ~0.5°C when using WOA18. The reviewer queried whether the NOAA-ERSST database introduces a bias to the SST anomalies we generate: our comparison indicates that on the whole there are minor offsets between the two products. However, regardless of which database is used, the main Benguela upwelling sites in the Pliocene continue to show SST anomalies which far exceed the PlioMIP2 model output (see Figure below). We therefore prefer to keep our reflection on the possible oceanographic causes of this data-model offset on page 11.

The reviewer comments that he finds lower SST anomalies at Site 1085 than our main Benguela sites when using WOA18. We note here that the +3°C anomaly he states is comparable with Site 1087, where the difference in the SST anomaly between NOAA-ERSST and WOA18 is also less than 0.5°C. Both Site 1085 and Site 1087 lie in the Southern Benguela region, which is today under greater influence from the Benguela Current (and potentially the Agulhas retroflection) than the main cells to the north (Sites 1081, 1082 and 1084; Wefer et al., 1998). It has also been shown that during the mid-Pliocene, coastal upwelling in the Southern Benguela region was enhanced compared to today (e.g. Petrick et al., 2018), which may account for the similarities between Sites 1085 and 1087, and their differences to those sites located in the central and northern upwelling region.

[Figure]

*Figure 1: comparison of SST anomalies for the proxy data, using NOAA-ERSST5 (left, as undertaken in the original submission) and World Ocean Atlas 2018 (right, Locarnini et al., 2018). Sites from the Benguela upwelling region are annotated.*

Literature cited:

Locarnini, R. A., A. V. Mishonov, O. K. Baranova, T. P. Boyer, M. M. Zweng, H. E. Garcia, J. R. Reagan, D. Seidov, K. Weathers, C. R. Paver, and I. Smolyar (2018) World Ocean Atlas 2018, Volume 1: Temperature. A. Mishonov Technical Ed.; NOAA Atlas NESDIS 81, 52 pp.

Petrick, B., McClymont, E. L., Littler, K., Rosell-Melé, A., Clarkson, M. O., Maslin, M., Röhl, U., Shevenell, A. E., and Pancost, R. D. (2018) Oceanographic and climatic evolution of the southeastern subtropical Atlantic over the last 3.5 Ma, Earth and Planetary Science Letters, 492, 12-21, https://doi.org/10.1016/j.epsl.2018.03.054.

Wefer, G., Berger, W.H., Richter, C., et al. (1998) Proc. ODP, Init. Repts., 175: College Station, TX (Ocean Drilling Program). doi:10.2973/odp.proc.ir.175.199

[revised manuscript text omitted]

**Supplemental Information**

**Age model update for ODP Sites 1090 and 806**

At ODP Site 1090, following initial publication of the SST data (Martínez-Garcia et al., 2010) an alternative orbitally-tuned age model was generated using *n*-alkane concentrations as a proxy for dust inputs, and an anticipated continuation of the Pleistocene relationship of high dust with high $\delta^{18}O$ i.e. during glacial stages (Martínez-Garcia et al., 2011). This *n*-alkane age model aligns KM5c with high *n*-alkane concentrations and low SSTs, whereas the reverse pattern might be expected (Figure S1). If the cold interval is re-aligned to KM4, SSTs during KM5c at ODP 1090 are elevated by 0.5°C (Figure S1). Given current stratigraphic information for ODP 1090 it is not possible to determine which of these scenarios is correct; thus, we present the SST anomalies according to the original age model, noting that there could be an additional increase in those anomalies of up to 0.5°C depending upon the choice of sample ages.

At ODP Site 806, uncertainty over age control resulted from the absence of an agreed splice across the multiple holes drilled by ODP. High-resolution benthic foraminifera $\delta^{18}O$ records were generated on Hole 806B (Bickert et al., 1993;Karas et al., 2009). Here we update the age model using the HMM-Stack Matlab code (Lin et al., 2014), which aligns to the Prob-stack (Ahn et al., 2017). Additionally, we created a modified meters composite depth (mcd). Using the depth scale generated by Karas et al., (2009) to account for core expansion, we amend Holes 806A and 806C to this depth scale (*Matlab code is provided as a supplement*). The KM5c interval is muted in Prob-stack in comparison to LR04 (Ahn et al., 2017). Given the variability in the Site 806 benthic $\delta^{18}O$ record (Figure 1), it is difficult to identify the KM5c interval and we rely on the probabilistic alignment of HMM-Match. If we tied the record to LR04 between M2 and KM2 and assumed a linear sedimentation rate, however, the age model in practice would be similar.

**Alkenone calibrations**

The majority of the alkenone-derived sea-surface temperature (SST) datasets included in the PlioVAR synthesis used the $U^{K}_{37}$' index, and applied the core-top calibration (60°S–60°N) by Müller et al. (1998) (hereafter Müller98; Tables S2 and S3). Several PlioVAR datasets were originally published using the laboratory culture calibration of *Emiliania huxleyi* by Prahl et al. (1988) (Table S3); these data were converted to Müller98 so that all sites used the same linear global calibration. The Bayesian $U^{K}_{37}$' calibration (BAYSPLINE) was then applied to all sites. Whilst the Müller98 calibration indicates mean annual SSTs for high latitudes, at sites >45°N (Pacific) and >48°N (Atlantic), and in the Mediterranean Sea, BAYSPLINE explicitly reconstructs seasonal SST (Tierney and Tingley, 2018).

Table S3 and Figure S2 compare the reconstructed SST anomalies for KM5c (relative to pre-industrial) for the 23 sites which provided alkenone data. In the mid- and high-latitudes, Müller98 tends to generate warmer SSTs compared to BAYSPLINE, with the difference ≤0.9 °C (Table S3). There is relatively little variability in the offset (± 0.15 °C) although that may reflect low sample numbers for some sites (Figure S2). In the low latitudes, where SSTs exceed ~24.5°C (applying Müller98), the non-linearity of the BAYSPLINE calibration has its biggest impact (Figure S2). For most low-latitude sites SSTs are ~1°C warmer using BAYSPLINE, but the difference can be as high as 1.67 °C ± 0.01°C (ODP 1143). The warmer low-latitudes in BAYSPLINE reduce the meridional temperature gradient, but both Müller98 and BAYSPLINE are consistent in showing enhanced warming at mid- and high-latitudes.

**Foraminifera Mg/Ca calibrations**

A range of foraminifera species, Mg/Ca-SST calibrations, and corrections for non-thermal impacts on Mg/Ca had been employed for the original published data (Table S4). We present the data as published, recognising the choices made by the original researchers in identifying the best approach for their site. The Bayesian calibration,
 BAYMAG, was then applied to all data following the settings detailed in the Methods.

Table S4 and Figure S3 compared the reconstructed SST anomalies for KM5c (relative to pre-industrial) for the 12 sites which provided foraminifera Mg/Ca data. A wide range of offsets is recorded, both positive and negative, and there is no clear pattern in terms of latitude or species.

880 **Table S1.** Sites used in the PlioVAR synthesis, their age constraints and SST proxies, can be accessed at https://pliovar.github.io/km5c.html.

**Table S2. Alkenone indices and temperature calibrations discussed in the text. [$C_{37:x}$] refers to the concentration of the $C_{37}$ alkenone with x unsaturations.**

| Alkenone index | Calibration to ocean temperature | Sample type; Interpretation | Calibration reference |
|---|---|---|---|
| $U^{K}_{37}$' = [$C_{37:2}$] / ([$C_{37:2}$] + [$C_{37:3}$]) | $U^{K}_{37}$' = 0.034T + 0.039 | *Emiliania huxleyi* cultures; Growth temperature | Prahl et al. (1998) |
| $U^{K}_{37}$' (as above) | $U^{K}_{37}$' = 0.033T + 0.044 | Core tops, 60°S to 60°N; Mean annual SST | Müuller et al. (1998) |
| $U^{K}_{37}$' (as above) | Bayesian calibration (BAYSPLINE) | Core tops, 60°S to 70°N Mean annual SST, except seasonal SST in high latitudes (>48°N) and Mediterranean | Tierney and Tingley (2018) |

885

**Table S3: The impact of applying two alkenone calibrations on the PlioVAR SST reconstructions for KM5c (3.195–3.215 Ma), sorted by basin and latitude (from N to S). All data were converted to the Müller et al. (1998) calibration prior to analysis. The recommended prior standard deviation scalar (pstd) of 10 was applied to all sites, excluding for high $U^{K}_{37}$' values where the more restrictive value of 5 was used, as recommended in the BAYSPLINE calibration (Tierney and Tingley, 2018).**

| Site | Original calibration | Original reference(s) | T difference (BAYSPLINE 50% level - Muller 98) |
|---|---|---|---|
| *Atlantic Ocean and Mediterranean Sea* | | | |
| 907 | Müller et al. (1998) | Herbert et al. (2016) | - 0.34 °C (*n* = 1) |
| 642 | Müller et al. (1998) | Bachem et al. (2016) | - 0.66 °C ± 0.02 °C |
| 982 | Prahl et al. (1998) | Herbert et al. (2016), Lawrence et al. (2009) | - 0.70 °C ± 0.01 °C |
| U1313 | Müller et al. (1998) | Naafs et al. (2010) | - 0.74 °C ± 0.01 °C |
| 607 | Prahl et al. (1998) | Lawrence et al. (2010) | - 0.74 °C ± 0.02 °C |
| 999 | Müller et al. (1998) Sonzogni et al. (1997) | Badger et al. (2013) Seki et al. (2010) | + 0.87 °C ± 0.16 °C *(BAYSPLINE pstd = 5)* |
| 662 | Müller et al. (1998) | Herbert et al. (2010) | +1.25 °C ± 0.08 °C *(BAYSPLINE pstd=5)* |
| U1387 | Müller et al. (1998) | Tzanova & Herbert (2015) | +0.34 °C ± 0.28 °C *(BAYSPLINE pstd=5)* |
| Punto Piccola | Müller et al. (1998) | Herbert et al. (2015) | -0.19 °C ± 0.08 °C *(BAYSPLINE pstd=5)* |
| 609 | Müller et al. (1998) | Lawrence and Woodard (2017) | -0.71 °C ± 0.02 °C |
| 625 | Müller et al. (1998) | Van der Weijst and Peterse (unpublished) | +0.92 °C ± 0.16 °C *(BAYSPLINE pstd=5)* |
| 1081 | Müller et al. (1998) | Rosell-Melée et al. (2014) | -0.47 °C (*n* = 1) |
| 1082 | Müller et al. (1998) | Etourneau et al. (2009) | -0.19 °C ± 0.15 °C |
| 1084 | Müller et al. (1998) | Rosell-Melée et al. (2014) | -0.51 °C ± 0.07 °C |
| 1087 | Müller et al. (1998) | Petrick et al. (2015) | -0.86 °C ± 0.01 °C |
| 1090 | Müller et al. (1998) | Martínez-Garcia et al. (2011;2010) | -0.39 °C ± 0.01 °C |
| *Pacific Ocean* | | | |
| 1143 | Müller et al. (1998) | Li et al. (2011) | +1.67 °C ± 0.01 °C *(BAYSPLINE pstd=5)* |
| U1417 | Müller et al. (1998) | Sáanchez-Montes et al. (2019) | -0.59 °C (*n* = 1) |
| 846 | Müller et al. (1998) | Lawrence et al. (2006) | -0.23 °C ± 0.09 °C *(BAYSPLINE pstd=5)* |
| U1337 | Müller et al. (1998) | Li et al. (2019) | +1.65 °C (*n* = 1) |
| 593 | Müller et al. (1998) | McClymont et al. (2016) | -0.70 °C ± 0.02 °C |
| 594 | Müller et al. (1998) | Cabellero-Gill et al. (2019) | -0.64 °C ± 0.01 °C |
| 1125 | Müller et al. (1998) | Cabellero-Gill et al. (2019) | -0.74 °C ± 0.01 °C |
| *Indian Ocean* | | | |

| 722 | Müller et al. (1998) | Herbert et al. (2010) | +0.91 °C ± 0.08 °C |
| | | | *(BAYSPLINE pstd=5)* |

**Table S4: Comparison of published Mg/Ca calibration and BAYMAG for PlioVAR SST reconstructions for KM5c (3.195–3.215 Ma), sorted by basin and latitude (from N to S). The original Mg/Ca SST calibrations (and any corrections) used in the published datasets are shown.**

| Site | Species | Original calibration | Original reference(s) | T difference (BAYMAG – published calibration) |
|------|---------|----------------------|------------------------|-----------------------------------------------|
| *Atlantic Ocean* | | | | |
| 609 | *G. bulloides* | Mashiotta et al. (1999) | Bartoli et al. (2005) | +0.80 °C ± 0.83 °C |
| U1313 | *G. bulloides* | Elderfield and Ganssen (2000) | Hennissen et al. (2014) | +4.75 °C ± 0.23 °C |
| 603 | *G. bulloides* | Elderfield and Ganssen (2000) | De Schepper et al. (2009) | +4.73 °C ± 0.74 °C |
| 999 | *T. sacculifer* | Nürnberg et al. (2000) | De Schepper et al. (2013) | -0.47 °C ± 0.09 °C |
| 959 | *T. sacculifer* | Dekens et al. (2002), which includes a dissolution correction, with Evans et al. (2016) Mg/Ca$_{sw}$ correction | Van der Weijst and Peterse (unpublished) | -4.12 °C ± 0.24 °C |
| 516 | *T. sacculifer* | Anand et al. (2003) | Karas et al., (2017) | +1.26 °C ± 0.24 °C |
| *Pacific Ocean* | | | | |
| 1143 | *G. ruber* | Dekens et al. (2002), which includes a dissolution correction, | Tian et al., (2006) | -0.57 °C ± 0.17 °C |
| 1241 | *T. sacculifer* | Nürnberg et al. (2000) | Groeneveld et al. (2006) | +2.73 °C ± 0.37 °C |
| 806 | *T. sacculifer* | Dekens et al. (2002) , which includes a dissolution correction, | Wara et al. (2005) | +1.46 °C ± 0.04 °C |
| *Indian Ocean* | | | | |
| 709 | *T. sacculifer* | Anand et al. (2003) with Regenberg et al. (2006) dissolution correction | Karas et al. (2011) | +0.29 °C (*n* = 1) |
| 214 | *T. sacculifer* | Anand et al. (2003) with Regenberg et al. (2006) dissolution correction | Karas et al. (2009) | -1.05 °C (*n* = 1) |
| 763 | *T. sacculifer* | Anand et al. (2003) with Regenberg et al. (2006) dissolution correction | Karas et al. (2011) | +1.34 °C (*n* = 1) |

[Figure]

**Figure S1: Age control for ODP 1090.** *n*-Alkane concentrations and SSTs plotted on the original age scale of Martínez-Garcia et al. (2011), whereby the Pleistocene relationship of high *n*-alkane concentrations during glacial stages was applied. The KM5c window adopted in the main text is indicated by the vertical yellow bar. An alternative alignment of the published KM5c *n*-alkane peak and SST minimum into KM4 or the final stages of KM5c (dashed lines) leads to an increase in KM5c SSTs of up to 0.5°C (grey box).

[Figure]

Figure S2: the impact of applying either the non-linear BAYSPLINE (Tierney and Tingley, 2018) or linear Müller et al. (1998) calibrations for the alkenone $U^{K}_{37}$' index for the KM5c interval. Temperature anomaly information is also provided in Table S3. Sites are ordered by latitude as shown in Figure 4 of the main text (594 at 46°S through to 907 at 69°N). Four sites contain only one data point for the KM5c interval (1081, 1337, 1417 and 907).

[Figure]

**Figure S3: impact of applying BAYMAG to original (published) Mg/Ca temperature calibrations. Temperature anomaly information is provided in Table S4. Sites are ordered by latitude as shown in Figure 4 of the main text (516 at -30°S to 609 at 50°N). Three sites contain only one data point for the KM5c interval (763, 214, 709).**

[Figure]

930 **Figure S4: The impact of the numbers of data points within KM5c (#sample) on the temporal variability of SST data (standard deviation; SD). For most sites, SD is <1ºC (and closer to 0-0.5 ºC).**

[Figure]

**Figure S54: impact of changing high/low latitude bands on meridional SST gradient calculations. The high-latitude box is expanded from >60°N/S (Figure 3) to include sites between 45-60°N/S, and the low-latitude box is restricted to 15°S-15°N. This adds a further 4 sites to the original 2 included in the high-latitude box, and removes the possible influence of the Benguela upwelling sites from the low-latitude SST calculations, given data-model mismatch (Figure 4). Although there is minimal change in the proxy data meridional T gradient anomaly (2.8°C here compared to 2.6°C in Figure 32), the data no longer agree with the PlioMIP2 models.**

**3.215 - 3.195 Ma**

[Figure]

**Figure S65: Absolute SSTs for each site, for modern (World Ocean Atlas, 2018 (Boyer et al., 2018)) and for KM5c (proxy data and models as for Figure 4). Grey shading represents the range of SSTs recorded at each latitude for WOA18 (the zonal mean is shown by the solid black line).**

945

[Figure]

[Figure]

[Figure]

**Figure S7: Impact of pre-industrial choice on the anomaly calculation. Top: ERSSTv5 (as shown in Figure 4 of the main text); middle: the anomalies using the nearest available core-top data (for alkenones) and the forward-modelled 'core-top' from BAYMAG (Tierney et al., 2019), which uses World Ocean Atlas SST data (Locarnini et al., 2013); bottom: the anomalies calculated against World Ocean Atlas 2018 (Locarnini et al., 2018). For site information see https://pliovar.github.io/km5c.html).**

**Supplement reference list**

Ahn, S., Khider, D., Lisiecki, L. E., and Lawrence, C. E.: A probabilistic Pliocene–Pleistocene stack of benthic $\delta^{18}O$ using a profile hidden Markov model, Dynamics and Statistics of the Climate System, 2, 10.1093/climsys/dzx002, 2017.

Anand, P., and Elderfield, H.: Calibration of Mg/Ca thermometry in planktonic foraminifera from a sediment trap time series, Paleoceanography, 18, 1050, doi:1010.1029/2002PA000846, 2003.

Bachem, P. E., Risebrobakken, B., and McClymont, E. L.: Sea surface temperature variability in the Norwegian Sea during the late Pliocene linked to subpolar gyre strength and radiative forcing, Earth and Planetary Science Letters, 446, 113-122, http://dx.doi.org/10.1016/j.epsl.2016.04.024, 2016.

Badger, M. P. S., Schmidt, D. N., Mackensen, A., and Pancost, R. D.: High resolution alkenone palaeobarometry indicates stable $p$CO$_2$ during the Pliocene (3.3 to 2.8 Ma), Proceedings of the Royal Society, A, 371, Article 20130094, 2013.

Bartoli, G., Sarnthein, M., Weinelt, M., Erlenkeuser, H., Garbe-Schönberg, D., and Lea, D. W.: Final closure of Panama and the onset of northern hemisphere glaciation, Earth and Planetary Science Letters, 237, 33-44, https://doi.org/10.1016/j.epsl.2005.06.020, 2005.

Bickert, T., Berger, W. H., Burke, S., Schmidt, H., and Wefer, G.: Late Quaternary stable isotope record of benthic foraminifers: Sites 805 and 806, Ontong Java Plateau, in: Proc. ODP, Sci. Results, 130, edited by: Berger, W. H., Kroenke, L. W., Mayer, L. A., and et al., College Station, TX (Ocean Drilling Program), 411–420. doi:410.2973/odp.proc.sr.2130.2025.1993, 1993.

Boyer, T. P., Baranova, O. K., Coleman, C., Garcia, H. E., Grodsky, A., Locarnini, R. A., Mishonov, A. V., O'Brien, T. D., Paver, C. R., Reagan, J. R., Seidov, D., Smolyar, I. V., Weathers, K., and Zweng, M. M.: World Ocean Database 2018, in preparation, https://www.nodc.noaa.gov/OC5/indprod.html, 2018.

Caballero-Gill, R. P., Herbert, T. D., and Dowsett, H. J.: 100-kyr Paced Climate Change in the Pliocene Warm Period, Southwest Pacific, Paleoceanography and Paleoclimatology, 34, 524-545, 10.1029/2018pa003496, 2019.

De Schepper, S., Head, M. J., and Groeneveld, J.: North Atlantic Current variability through marine isotope stage M2 (circa 3.3. Ma) during the mid-Pliocene, Paleoceanography, 24, PA4206, doi:4210.1029/2008PA001725, 2009.

De Schepper, S., Groeneveld, J., Naafs, B. D. A., Van Renterghem, C., Hennissen, J., Head, M. J., Louwye, S., and Fabian, K.: Northern Hemisphere Glaciation during the Globally Warm Early Late Pliocene, PLoS ONE, 8, e81508. doi:81510.81371/journal.pone.0081508, 2013.

Dekens, P. S., Lea, D. W., Pak, D. K., and Spero, H. J.: Core top calibration of Mg/Ca in tropical foraminifera: Refining paleotemperature estimation, Geochemistry Geophysics Geosystems, 3, 1022, doi:1010.1029/2001GC000200. , 2002.

Elderfield, H., and Ganssen, G.: Past temperature and δ18O of surface ocean waters inferred from foraminiferal Mg/Ca ratios, Nature, 405, 442-445, 10.1038/35013033, 2000.

Etourneau, J., Martinez, P., Blanz, T., and Schneider, R.: Pliocene-Pleistocene variability of upwelling activity, productivity, and nutrient cycling in the Benguela region, Geology, 37, 871-874, 10.1130/g25733a.1, 2009.

Evans, D., Brierley, C., Raymo, M. E., Erez, J., and Müller, W.: Planktic foraminifera shell chemistry response to seawater chemistry: Pliocene–Pleistocene seawater Mg/Ca, temperature and sea level change, Earth and Planetary Science Letters, 438, 139-148, https://doi.org/10.1016/j.epsl.2016.01.013, 2016.

Groeneveld, J., Steph, S., Tiedemann, R., Garbe-Schönberg, C., Nürnberg, D., and Sturm, A.: Pliocene mixed-layer oceanography for Site 1241, using combined Mg/Ca and $\delta^{18}O$ analyses of Globigerinoides sacculifer, Proceedings of the Ocean Drilling Program: Scientific Results, 202, 1-27, 2006.

Hennissen, J. A. I., Head, M. J., De Schepper, S., and Groeneveld, J.: Palynological evidence for a southward shift of the North Atlantic Current at ~2.6 Ma during the intensification of late Cenozoic Northern Hemisphere glaciation, Paleoceanography, 29, 564-580, 10.1002/2013pa002543, 2014.

Herbert, T. D., Peterson, L. C., Lawrence, K. T., and Liu, Z.: Tropical Ocean Temperatures Over the Past 3.5 Million Years, Science, 328, 1530-1534, 10.1126/science.1185435, 2010.

Herbert, T. D., Ng, G., and Cleaveland Peterson, L.: Evolution of Mediterranean sea surface temperatures 3.5–1.5 Ma: Regional and hemispheric influences, Earth and Planetary Science Letters, 409, 307-318, http://dx.doi.org/10.1016/j.epsl.2014.10.006, 2015.

Herbert, T. D., Lawrence, K. T., Tzanova, A., Peterson, L. C., Caballero-Gill, R., and Kelly, C. S.: Late Miocene global cooling and the rise of modern ecosystems, Nature Geoscience, 9, 843-847, doi:810.1038/ngeo2813, 2016.

Karas, C., Nurnberg, D., Gupta, A. K., Tiedemann, R., Mohan, K., and Bickert, T.: Mid-Pliocene climate change amplified by a switch in Indonesian subsurface throughflow, Nature Geoscience, 2, 434-438, http://www.nature.com/ngeo/journal/v2/n6/suppinfo/ngeo520_S1.html, 2009.

Karas, C., Nürnberg, D., Tiedemann, R., and Garbe-Schönberg, D.: Pliocene Indonesian Throughflow and Leeuwin Current dynamics: Implications for Indian Ocean polar heat flux, Paleoceanography, 26, PA2217, 10.1029/2010pa001949, 2011.

Karas, C., Nürnberg, D., Bahr, A., Groeneveld, J., Herrle, J. O., Tiedemann, R., and deMenocal, P. B.: Pliocene oceanic seaways and global climate, Scientific Reports, 7, 39842, 10.1038/srep39842, 2017.

Lawrence, K. T., Liu, Z., and Herbert, T. D.: Evolution of the Eastern Tropical Pacific Through Plio-Pleistocene Glaciation, Science, 312, 79-83, 2006.

Lawrence, K. T., Herbert, T. D., Brown, C. M., Raymo, M. E., and Haywood, A. M.: High-amplitude variations in North Atlantic sea surface temperature during the early Pliocene warm period, Paleoceanography, 24, 2009.

Lawrence, K. T., Sosdian, S., White, H. E., and Rosenthal, Y.: North Atlantic climate evolution through the Plio-Pleistocene climate transitions, Earth and Planetary Science Letters, 300, 329-342, https://doi.org/10.1016/j.epsl.2010.10.013, 2010.

Lawrence, K. T., and Woodard, S. C.: Past sea surface temperatures as measured by different proxies—A cautionary tale from the late Pliocene, Paleoceanography, 32, 318-324, 10.1002/2017pa003101, 2017.

Li, L., Li, Q., Tian, J., Wang, P., Wang, H., and Liu, Z.: A 4-Ma record of thermal evolution in the tropical western Pacific and its implications on climate change, Earth and Planetary Science Letters, 309, 10-20, 2011.

Lin, L., Khider, D., Lisiecki, L. E., and Lawrence, C. E.: Probabilistic sequence alignment of stratigraphic records, Paleoceanography, 29, 976–989, 2014.

Liu, J., Tian, J., Liu, Z., Herbert, T. D., Fedorov, A. V., and Lyle, M.: Eastern equatorial Pacific cold tongue evolution since the late Miocene linked to extratropical climate, Science Advances, 5, eaau6060, 10.1126/sciadv.aau6060, 2019.

1030    Locarnini, R. A., A. V. Mishonov, O. K. Baranova, T. P. Boyer, M. M. Zweng, H. E. Garcia, J. R. Reagan, D. Seidov, K. Weathers, C. R. Paver, and I. Smolyar (2018) World Ocean Atlas 2018, Volume 1: Temperature. A. Mishonov Technical Ed.; NOAA Atlas NESDIS 81, 52 pp.

Martinez-Garcia, A., Rosell-Mele, A., McClymont, E. L., Gersonde, R., and Haug, G. H.: Subpolar Link to the Emergence of the Modern Equatorial Pacific Cold Tongue, Science, 328, 1550-1553, 10.1126/science.1184480, 2010.

1035    Martinez-Garcia, A., Rosell-Mele, A., Jaccard, S. L., Geibert, W., Sigman, D. M., and Haug, G. H.: Southern Ocean dust-climate coupling over the past four million years, Nature, 476, 312-315, 2011.

Mashiotta, T. A., Lea, D. W., and Spero, H. J.: Glacial–interglacial changes in Subantarctic sea surface temperature and δ18O-water using foraminiferal Mg, Earth and Planetary Science Letters, 170, 417-432, https://doi.org/10.1016/S0012-821X(99)00116-8, 1999.

1040    McClymont, E. L., Elmore, A. C., Kender, S., Leng, M. J., Greaves, M., and Elderfield, H.: Pliocene-Pleistocene evolution of sea surface and intermediate water temperatures from the Southwest Pacific, Paleoceanography, PA002954, 2016.

Müller, P. J., Kirst, G., Ruhland, G., Storch, I. V., and Rosell-Melé, A.: Calibration of the alkenone paleotemperature index $U^k_{37}{}'$ based on core-tops from the eastern South Atlantic and the global ocean (60°N–60°S), Geochimica et Cosmochimica Acta, 62, 1757–1772, 1998.

1045    Naafs, B. D. A., Stein, R., Hefter, J., Khélifi, N., De Schepper, S., and Haug, G. H.: Late Pliocene changes in the North Atlantic Current, Earth and Planetary Science Letters, 298, 434-442, http://dx.doi.org/10.1016/j.epsl.2010.08.023, 2010.

Nürnberg, D., Müller, A., and Schneider, R. R.: Paleo-sea surface temperature calculations in the equatorial east Atlantic from Mg/Ca ratios in planktic foraminifera: A comparison to sea surface temperature estimates from U37K′, oxygen isotopes, and foraminiferal transfer function, Paleoceanography, 15, 124-134, 10.1029/1999pa000370, 2000.

1050    Petrick, B., McClymont, E. L., Felder, S., Rueda, G., Leng, M. J., and Rosell-Melé, A.: Late Pliocene upwelling in the Southern Benguela region, Palaeogeography, Palaeoclimatology, Palaeoecology, 429, 62-71, http://dx.doi.org/10.1016/j.palaeo.2015.03.042, 2015.

Prahl, F. G., Muehlhausen, L. A., and Zahnle, D. I.: Further evaluation of long-chain alkenones as indicators of paleoceanographic conditions, Geochimica et Cosmochimica Acta, 52, 2303-2310, 1988.

1055    Regenberg, M., Nürnberg, D., Steph, S., Groeneveld, J., Garbe-Schönberg, D., Tiedemann, R., and Dullo, W.-C.: Assessing the effect of dissolution on planktonic foraminiferal Mg/Ca ratios: Evidence from Caribbean core tops, Geochemistry Geophysics Geosystems, 7, Q07P15, doi:10.1029/2005GC001019, 2006.

Rosell-Melé, A., Martínez-Garcia, A., and McClymont, E. L.: Persistent warmth across the Benguela upwelling system during the Pliocene epoch, Earth and Planetary Science Letters, 386, 10-20, http://dx.doi.org/10.1016/j.epsl.2013.10.041, 1060    2014.

Sánchez-Montes, M. L., McClymont, E. L., Lloyd, J. M., Müller, J., Cowan, E. A., and Zorzi, C.: Late Pliocene Cordilleran Ice Sheet development with warm Northeast Pacific sea surface temperatures, Clim. Past Discuss. (accepted), 2019, 1-23, 10.5194/cp-2019-29, 2019.

Seki, O., Foster, G. L., Schmidt, D. N., Mackensen, A., Kawamura, K., and Pancost, R. D.: Alkenone and boron-based
1065    Pliocene pCO2 records, Earth and Planetary Science Letters, 292, 201-211, 2010.

Tian, J., Pak, D. K., Wang, P., Lea, D., Cheng, X., and Zhao, Q.: Late Pliocene monsoon linkage in the tropical South China
Sea, Earth and Planetary Science Letters, 252, 72-81, https://doi.org/10.1016/j.epsl.2006.09.028, 2006.

Tierney, J. E., and Tingley, M. P.: BAYSPLINE: A New Calibration for the Alkenone Paleothermometer, 33, 281-301,
10.1002/2017pa003201, 2018.

1070    Tierney, J. E., Malevich, S. B., Gray, W., Vetter, L., and Thirumalai, K.: Bayesian calibration of the Mg/Ca
paleothermometer in planktic foraminifera, Paleoceanography and Paleoclimatology, 31,
https://doi.org/10.1029/2019PA003744, 10.1029/2019pa003744, 2019.

Tzanova, A., and Herbert, T. D.: Regional and global significance of Pliocene sea surface temperatures from the Gulf of
Cadiz (Site U1387) and the Mediterranean, Global and Planetary Change, 133, 371-377,
1075    https://doi.org/10.1016/j.gloplacha.2015.07.001, 2015.

Wara, M. W., Ravelo, A. C., and Delaney, M. L.: Permanent El Nino-Like Conditions During the Pliocene Warm Period,
Science, 309, 758-761, 2005.

---

## Author Response (AR2)

**CP-2019-161** *Lessons from a high CO2 world: an ocean view from ~3 million years ago*

**Reply to Editor**

This document addresses the concerns of the Editor on our revised manuscript. We also include the "tracked changes" manuscript text. No changes have been made to the Supplement text.

We thank the Editor for the remaining comments, which are addressed here and shown as tracked changes in the document below:

*Upon final reading of the manuscript I noticed 2 areas, that I feel need addressing before the manuscript can be published. These are minor and the second I think is a result of some changes you made to the manuscript in the most recent submission.*

*1. Please reference the underlying SST data in Figure 2. At present, the figure caption only makes reference to the the PlioVAR sites.*

Thank you for flagging this, we had omitted to cite our source data. The PlioVAR sites have been overlaid on the World Ocean Atlas (2018) mean annual SSTs. We have cited this source in the caption.

We also felt that we should highlight that the compiled information on sites and proxies is included in the Supplement file and in the Pangaea archive. Our caption has been edited accordingly:

> **Figure 2: Locations of sites used in the synthesis, overlain on mean annual SST data from the World Ocean Atlas 2018 (Locarnini et al., 2018). A full list of the data sources and proxies applied per site are available in Table S3 ($U^K_{37}$') and Table S4 (Mg/Ca), and can be accessed at https://pliovar.github.io/km5c.html. The combined PlioVAR proxy data and its sources are also archived at Pangaea: https://doi.pangaea.de/10.1594/PANGAEA.911847.**

*2. One line 282 of the revised submission, you state that: "Our proxy-based mean global SST is larger than most PlioMIP models (Figure 3); because air temperature increases are larger over the land than over the ocean (Haywood et al...)"...*

*Can you specify which 'proxy-based mean global SST you are refering to? Proxy data (all)?*

To keep consistency with the phrasing we are using elsewhere in the manuscript we have revised this statement (lines 309-310 below):

> **Our proxy-based mean global SST anomaly is larger than in most PlioMIP2 models when we use alkenones-only or alkenones and Mg/Ca combined (Figure 3), ...**

*Also, the statement regardig temperature increases over land being larger than over the ocean seems out of place here in reference to Figure 3, which I assume is depicting a global mean SST for the models as well as the data. Whilst the statement is true, I don't think it is a justification of the differences between models and data in Figure 3. Please can you have a look at lines 282-285 and consider rephrasing them.*

In this comment we were not seeking to explain the difference between the data and models in Figure 3 (since both are SST), but to flag that since our SST proxy reconstructions consistently gave values which were higher than the model SST data, then a larger warming in global annual surface air temperature might also be expected. The sentences which preceded this one were discussing other modelling studies which make comparisons between Pliocene and projected global annual surface air temperatures; we had wanted to flag here that we are showing SST anomalies (not surface air temperatures) and that the latter would likely be larger. However, we recognise that this became unclear in the text and have revised accordingly (lines 309-313 below):

[revised manuscript text omitted]

**Supplemental Information**

**Age model update for ODP Sites 1090 and 806**

At ODP Site 1090, following initial publication of the SST data (Martínez-Garcia et al., 2010) an alternative orbitally-tuned age model was generated using *n*-alkane concentrations as a proxy for dust inputs, and an anticipated continuation of the Pleistocene relationship of high dust with high $\delta^{18}O$ i.e. during glacial stages (Martínez-Garcia et al., 2011). This *n*-alkane age model aligns KM5c with high *n*-alkane concentrations and low SSTs, whereas the reverse pattern might be expected (Figure S1). If the cold interval is re-aligned to KM4, SSTs during KM5c at ODP 1090 are elevated by 0.5°C (Figure S1). Given current stratigraphic information for ODP 1090 it is not possible to determine which of these scenarios is correct; thus, we present the SST anomalies according to the original age model, noting that there could be an additional increase in those anomalies of up to 0.5°C depending upon the choice of sample ages.

At ODP Site 806, uncertainty over age control resulted from the absence of an agreed splice across the multiple holes drilled by ODP. High-resolution benthic foraminifera $\delta^{18}O$ records were generated on Hole 806B (Bickert et al., 1993;Karas et al., 2009). Here we update the age model using the HMM-Stack Matlab code (Lin et al., 2014), which aligns to the Prob-stack (Ahn et al., 2017). Additionally, we created a modified meters composite depth (mcd). Using the depth scale generated by Karas et al., (2009) to account for core expansion, we amend Holes 806A and 806C to this depth scale (*Matlab code is provided as a supplement*). The KM5c interval is muted in Prob-stack in comparison to LR04 (Ahn et al., 2017). Given the variability in the Site 806 benthic $\delta^{18}O$ record (Figure 1), it is difficult to identify the KM5c interval and we rely on the probabilistic alignment of HMM-Match. If we tied the record to LR04 between M2 and KM2 and assumed a linear sedimentation rate, however, the age model in practice would be similar.

**Alkenone calibrations**

The majority of the alkenone-derived sea-surface temperature (SST) datasets included in the PlioVAR synthesis used the $U^{K}_{37}$' index, and applied the core-top calibration (60°S–60°N) by Müller et al. (1998) (hereafter Müller98; Tables S2 and S3). Several PlioVAR datasets were originally published using the laboratory culture calibration of *Emiliania huxleyi* by Prahl et al. (1988) (Table S3); these data were converted to Müller98 so that all sites used the same linear global calibration. The Bayesian $U^{K}_{37}$' calibration (BAYSPLINE) was then applied to all sites. Whilst the Müller98 calibration indicates mean annual SSTs for high latitudes, at sites >45°N (Pacific) and >48°N (Atlantic), and in the Mediterranean Sea, BAYSPLINE explicitly reconstructs seasonal SST (Tierney and Tingley, 2018).

Table S3 and Figure S2 compare the reconstructed SST anomalies for KM5c (relative to pre-industrial) for the 23 sites which provided alkenone data. In the mid- and high-latitudes, Müller98 tends to generate warmer SSTs compared to BAYSPLINE, with the difference ≤0.9 °C (Table S3). There is relatively little variability in the offset (± 0.15 °C) although that may reflect low sample numbers for some sites (Figure S2). In the low latitudes, where SSTs exceed ~24.5°C (applying Müller98), the non-linearity of the BAYSPLINE calibration has its biggest impact (Figure S2). For most low-latitude sites SSTs are ~1°C warmer using BAYSPLINE, but the difference can be as high as 1.67 °C ± 0.01°C (ODP 1143). The warmer low-latitudes in BAYSPLINE reduce the meridional temperature gradient, but both Müller98 and BAYSPLINE are consistent in showing enhanced warming at mid- and high-latitudes.

**Foraminifera Mg/Ca calibrations**

A range of foraminifera species, Mg/Ca-SST calibrations, and corrections for non-thermal impacts on Mg/Ca had been employed for the original published data (Table S4). We present the data as

945     published, recognising the choices made by the original researchers in identifying the best approach for their site. The Bayesian calibration, BAYMAG, was then applied to all data following the settings detailed in the Methods.

    Table S4 and Figure S3 compared the reconstructed SST anomalies for KM5c (relative to pre-industrial) for the 12 sites which provided foraminifera Mg/Ca data. A wide range of offsets is
950     recorded, both positive and negative, and there is no clear pattern in terms of latitude or species.

**Table S1.** Sites used in the PlioVAR synthesis, their age constraints and SST proxies, can be accessed at https://pliovar.github.io/km5c.html.

955    **Table S2. Alkenone indices and temperature calibrations discussed in the text. [C$_{37:x}$] refers to the concentration of the C$_{37}$ alkenone with x unsaturations.**

| Alkenone index | Calibration to ocean temperature | Sample type; Interpretation | Calibration reference |
|---|---|---|---|
| U$^K_{37}$' = [C$_{37:2}$] / ([C$_{37:2}$] + [C$_{37:3}$]) | U$^K_{37}$' = 0.034T + 0.039 | *Emiliania huxleyi* cultures; Growth temperature | Prahl et al. (1998) |
| U$^K_{37}$' (as above) | U$^K_{37}$' = 0.033T + 0.044 | Core tops, 60°S to 60°N; Mean annual SST | Müller et al. (1998) |
| U$^K_{37}$' (as above) | Bayesian calibration (BAYSPLINE) | Core tops, 60°S to 70°N Mean annual SST, except seasonal SST in high latitudes (>48°N) and Mediterranean | Tierney and Tingley (2018) |

**Table S3: The impact of applying two alkenone calibrations on the PlioVAR SST reconstructions for KM5c (3.195–3.215 Ma), sorted by basin and latitude (from N to S). All data were converted to the Müller et al. (1998) calibration prior to analysis. The recommended prior standard deviation scalar (pstd) of 10 was applied to all sites, excluding for high $U^{K}_{37}$' values where the more restrictive value of 5 was used, as recommended in the BAYSPLINE calibration (Tierney and Tingley, 2018).**

| Site | Original calibration | Original reference(s) | T difference (BAYSPLINE 50% level - Muller 98) |
|---|---|---|---|
| *Atlantic Ocean and Mediterranean Sea* | | | |
| 907 | Müller et al. (1998) | Herbert et al. (2016) | - 0.34 °C (*n* = 1) |
| 642 | Müller et al. (1998) | Bachem et al. (2016) | - 0.66 °C ± 0.02 °C |
| 982 | Prahl et al. (1998) | Herbert et al. (2016), Lawrence et al. (2009) | - 0.70 °C ± 0.01 °C |
| U1313 | Müller et al. (1998) | Naafs et al. (2010) | - 0.74 °C ± 0.01 °C |
| 607 | Prahl et al. (1998) | Lawrence et al. (2010) | - 0.74 °C ± 0.02 °C |
| 999 | Müller et al. (1998) Sonzogni et al. (1997) | Badger et al. (2013) Seki et al. (2010) | + 0.87 °C ± 0.16 °C *(BAYSPLINE pstd = 5)* |
| 662 | Müller et al. (1998) | Herbert et al. (2010) | +1.25 °C ± 0.08 °C *(BAYSPLINE pstd=5)* |
| U1387 | Müller et al. (1998) | Tzanova & Herbert (2015) | +0.34 °C ± 0.28 °C *(BAYSPLINE pstd=5)* |
| Punto Piccola | Müller et al. (1998) | Herbert et al. (2015) | -0.19 °C ± 0.08 °C *(BAYSPLINE pstd=5)* |
| 609 | Müller et al. (1998) | Lawrence and Woodard (2017) | -0.71 °C ± 0.02 °C |
| 625 | Müller et al. (1998) | Van der Weijst and Peterse (unpublished) | +0.92 °C ± 0.16 °C *(BAYSPLINE pstd=5)* |
| 1081 | Müller et al. (1998) | Rosell-Melé et al. (2014) | -0.47 °C (*n* = 1) |
| 1082 | Müller et al. (1998) | Etourneau et al. (2009) | -0.19 °C ± 0.15 °C |
| 1084 | Müller et al. (1998) | Rosell-Melé et al. (2014) | -0.51 °C ± 0.07 °C |
| 1087 | Müller et al. (1998) | Petrick et al. (2015) | -0.86 °C ± 0.01 °C |
| 1090 | Müller et al. (1998) | Martínez-Garcia et al. (2011;2010) | -0.39 °C ± 0.01 °C |
| *Pacific Ocean* | | | |
| 1143 | Müller et al. (1998) | Li et al. (2011) | +1.67 °C ± 0.01 °C *(BAYSPLINE pstd=5)* |
| U1417 | Müller et al. (1998) | Sánchez-Montes et al. (2019) | -0.59 °C (*n* = 1) |
| 846 | Müller et al. (1998) | Lawrence et al. (2006) | -0.23 °C ± 0.09 °C *(BAYSPLINE pstd=5)* |
| U1337 | Müller et al. (1998) | Li et al. (2019) | +1.65 °C (*n* = 1) |
| 593 | Müller et al. (1998) | McClymont et al. (2016) | -0.70 °C ± 0.02 °C |
| 594 | Müller et al. (1998) | Cabellero-Gill et al. (2019) | -0.64 °C ± 0.01 °C |
| 1125 | Müller et al. (1998) | Cabellero-Gill et al. (2019) | -0.74 °C ± 0.01 °C |
| *Indian Ocean* | | | |
| 722 | Müller et al. (1998) | Herbert et al. (2010) | +0.91 °C ± 0.08 °C *(BAYSPLINE pstd=5)* |

**Table S4: Comparison of published Mg/Ca calibration and BAYMAG for PlioVAR SST reconstructions for KM5c (3.195–3.215 Ma), sorted by basin and latitude (from N to S). The original Mg/Ca SST calibrations (and any corrections) used in the published datasets are shown.**

970

| Site | Species | Original calibration | Original reference(s) | T difference (BAYMAG – published calibration) |
|------|---------|---------------------|----------------------|-----------------------------------------------|
| *Atlantic Ocean* | | | | |
| 609 | *G. bulloides* | Mashiotta et al. (1999) | Bartoli et al. (2005) | +0.80 °C ± 0.83 °C |
| U1313 | *G. bulloides* | Elderfield and Ganssen (2000) | Hennissen et al. (2014) | +4.75 °C ± 0.23 °C |
| 603 | *G. bulloides* | Elderfield and Ganssen (2000) | De Schepper et al. (2009) | +4.73 °C ± 0.74 °C |
| 999 | *T. sacculifer* | Nürnberg et al. (2000) | De Schepper et al. (2013) | -0.47 °C ± 0.09 °C |
| 959 | *T. sacculifer* | Dekens et al. (2002), which includes a dissolution correction, with Evans et al. (2016) Mg/Ca$_{sw}$ correction | Van der Weijst and Peterse (unpublished) | -4.12 °C ± 0.24 °C |
| 516 | *T. sacculifer* | Anand et al. (2003) | Karas et al., (2017) | +1.26 °C ± 0.24 °C |
| *Pacific Ocean* | | | | |
| 1143 | *G. ruber* | Dekens et al. (2002), which includes a dissolution correction, | Tian et al., (2006) | -0.57 °C ± 0.17 °C |
| 1241 | *T. sacculifer* | Nürnberg et al. (2000) | Groeneveld et al. (2006) | +2.73 °C ± 0.37 °C |
| 806 | *T. sacculifer* | Dekens et al. (2002) , which includes a dissolution correction, | Wara et al. (2005) | +1.46 °C ± 0.04 °C |
| *Indian Ocean* | | | | |
| 709 | *T. sacculifer* | Anand et al. (2003) with Regenberg et al. (2006) dissolution correction | Karas et al. (2011) | +0.29 °C (*n* = 1) |
| 214 | *T. sacculifer* | Anand et al. (2003) with Regenberg et al. (2006) dissolution correction | Karas et al. (2009) | -1.05 °C (*n* = 1) |
| 763 | *T. sacculifer* | Anand et al. (2003) with Regenberg et al. (2006) dissolution correction | Karas et al. (2011) | +1.34 °C (*n* = 1) |

[Figure]

**Figure S1: Age control for ODP 1090.** *n*-Alkane concentrations and SSTs plotted on the original age scale of Martínez-Garcia et al. (2011), whereby the Pleistocene relationship of high *n*-alkane concentrations during glacial stages was applied. The KM5c window adopted in the main text is indicated by the vertical yellow bar. An alternative alignment of the published KM5c *n*-alkane peak and SST minimum into KM4 or the final stages of KM5c (dashed lines) leads to an increase in KM5c SSTs of up to 0.5°C (grey box).

[Figure]

**Figure S2: the impact of applying either the non-linear BAYSPLINE (Tierney and Tingley, 2018) or linear Müller et al. (1998) calibrations for the alkenone $U^{K}_{37}{}'$ index for the KM5c interval. Temperature anomaly information is also provided in Table S3. Sites are ordered by latitude as shown in Figure 4 of the main text (594 at 46°S through to 907 at 69°N). Four sites contain only one data point for the KM5c interval (1081, 1337, 1417 and 907).**

[Figure]

**Figure S3: impact of applying BAYMAG to original (published) Mg/Ca temperature calibrations.** Temperature anomaly information is provided in Table S4. Sites are ordered by latitude as shown in Figure 4 of the main text (516 at -30°S to 609 at 50°N). Three sites contain only one data point for the KM5c interval (763, 214, 709).

1000

[Figure]

1005

**Figure S4: The impact of the numbers of data points within KM5c (#sample) on the temporal variability of SST data (standard deviation; SD). For most sites, SD is <1ºC (and closer to 0-0.5 ºC).**

[Figure]

1010

**Figure S5: impact of changing high/low latitude bands on meridional SST gradient calculations. The high-latitude box is expanded from >60°N/S (Figure 3) to include sites between 45-60°N/S, and the low-latitude box is restricted to 15°S-15°N. This adds a further 4 sites to the original 2 included in the high-latitude box, and removes the possible influence of the Benguela upwelling sites from the low-latitude SST**
1015 **calculations, given data-model mismatch (Figure 4). Although there is minimal change in the proxy data meridional T gradient anomaly (2.8°C here compared to 2.6°C in Figure 3), the data no longer agree with the PlioMIP2 models.**

**3.215 - 3.195 Ma**

[Figure]

1020

**Figure S6: Absolute SSTs for each site, for modern (World Ocean Atlas, 2018 (Boyer et al., 2018)) and for KM5c (proxy data and models as for Figure 4). Grey shading represents the range of SSTs recorded at each latitude for WOA18 (the zonal mean is shown by the solid black line).**

1025

[Figure]

[Figure]

[Figure]

**Figure S7: Impact of pre-industrial choice on the anomaly calculation. Top: ERSSTv5 (as shown in Figure 4 of the main text); middle: the anomalies using the nearest available core-top data (for alkenones) and the forward-modelled 'core-top' from BAYMAG (Tierney et al., 2019), which uses World Ocean Atlas SST data (Locarnini et al., 2013); bottom: the anomalies calculated against World Ocean Atlas 2018 (Locarnini et al., 2018). For site information see https://pliovar.github.io/km5c.html).**

**Supplement reference list**

Ahn, S., Khider, D., Lisiecki, L. E., and Lawrence, C. E.: A probabilistic Pliocene–Pleistocene stack of benthic $\delta^{18}O$ using a profile hidden Markov model, Dynamics and Statistics of the Climate System, 2, 10.1093/climsys/dzx002, 2017.

1045    Anand, P., and Elderfield, H.: Calibration of Mg/Ca thermometry in planktonic foraminifera from a sediment trap time series, Paleoceanography, 18, 1050, doi:1010.1029/2002PA000846, 2003.

Bachem, P. E., Risebrobakken, B., and McClymont, E. L.: Sea surface temperature variability in the Norwegian Sea during the late Pliocene linked to subpolar gyre strength and radiative forcing, Earth and Planetary Science Letters, 446, 113-122, http://dx.doi.org/10.1016/j.epsl.2016.04.024, 2016.

1050    Badger, M. P. S., Schmidt, D. N., Mackensen, A., and Pancost, R. D.: High resolution alkenone palaeobarometry indicates stable $p$CO$_2$ during the Pliocene (3.3 to 2.8 Ma), Proceedings of the Royal Society, A, 371, Article 20130094, 2013.

Bartoli, G., Sarnthein, M., Weinelt, M., Erlenkeuser, H., Garbe-Schönberg, D., and Lea, D. W.: Final closure of Panama and the onset of northern hemisphere glaciation, Earth and Planetary Science Letters, 237, 33-44,
1055    https://doi.org/10.1016/j.epsl.2005.06.020, 2005.

Bickert, T., Berger, W. H., Burke, S., Schmidt, H., and Wefer, G.: Late Quaternary stable isotope record of benthic foraminifers: Sites 805 and 806, Ontong Java Plateau, in: Proc. ODP, Sci. Results, 130, edited by: Berger, W. H., Kroenke, L. W., Mayer, L. A., and et al., College Station, TX (Ocean Drilling Program), 411–420. doi:410.2973/odp.proc.sr.2130.2025.1993, 1993.

1060    Boyer, T. P., Baranova, O. K., Coleman, C., Garcia, H. E., Grodsky, A., Locarnini, R. A., Mishonov, A. V., O'Brien, T. D., Paver, C. R., Reagan, J. R., Seidov, D., Smolyar, I. V., Weathers, K., and Zweng, M. M.: World Ocean Database 2018, in preparation, https://www.nodc.noaa.gov/OC5/indprod.html, 2018.

Caballero-Gill, R. P., Herbert, T. D., and Dowsett, H. J.: 100-kyr Paced Climate Change in the Pliocene Warm Period, Southwest Pacific, Paleoceanography and Paleoclimatology, 34, 524-545, 10.1029/2018pa003496,
1065    2019.

De Schepper, S., Head, M. J., and Groeneveld, J.: North Atlantic Current variability through marine isotope stage M2 (circa 3.3. Ma) during the mid-Pliocene, Paleoceanography, 24, PA4206, doi:4210.1029/2008PA001725, 2009.

De Schepper, S., Groeneveld, J., Naafs, B. D. A., Van Renterghem, C., Hennissen, J., Head, M. J., Louwye, S.,
1070    and Fabian, K.: Northern Hemisphere Glaciation during the Globally Warm Early Late Pliocene, PLoS ONE, 8, e81508. doi:81510.81371/journal.pone.0081508, 2013.

Dekens, P. S., Lea, D. W., Pak, D. K., and Spero, H. J.: Core top calibration of Mg/Ca in tropical foraminifera: Refining paleotemperature estimation, Geochemistry Geophysics Geosystems, 3, 1022, doi:1010.1029/2001GC000200. , 2002.

1075    Elderfield, H., and Ganssen, G.: Past temperature and δ18O of surface ocean waters inferred from foraminiferal Mg/Ca ratios, Nature, 405, 442-445, 10.1038/35013033, 2000.

Etourneau, J., Martinez, P., Blanz, T., and Schneider, R.: Pliocene-Pleistocene variability of upwelling activity, productivity, and nutrient cycling in the Benguela region, Geology, 37, 871-874, 10.1130/g25733a.1, 2009.

Evans, D., Brierley, C., Raymo, M. E., Erez, J., and Müller, W.: Planktic foraminifera shell chemistry response
1080    to seawater chemistry: Pliocene–Pleistocene seawater Mg/Ca, temperature and sea level change, Earth and Planetary Science Letters, 438, 139-148, https://doi.org/10.1016/j.epsl.2016.01.013, 2016.

Groeneveld, J., Steph, S., Tiedemann, R., Garbe-Schönberg, C., Nürnberg, D., and Sturm, A.: Pliocene mixed-layer oceanography for Site 1241, using combined Mg/Ca and $\delta^{18}O$ analyses of Globigerinoides sacculifer, Proceedings of the Ocean Drilling Program: Scientific Results, 202, 1-27, 2006.

1085 Hennissen, J. A. I., Head, M. J., De Schepper, S., and Groeneveld, J.: Palynological evidence for a southward shift of the North Atlantic Current at ~2.6 Ma during the intensification of late Cenozoic Northern Hemisphere glaciation, Paleoceanography, 29, 564-580, 10.1002/2013pa002543, 2014.

Herbert, T. D., Peterson, L. C., Lawrence, K. T., and Liu, Z.: Tropical Ocean Temperatures Over the Past 3.5 Million Years, Science, 328, 1530-1534, 10.1126/science.1185435, 2010.

1090 Herbert, T. D., Ng, G., and Cleaveland Peterson, L.: Evolution of Mediterranean sea surface temperatures 3.5–1.5 Ma: Regional and hemispheric influences, Earth and Planetary Science Letters, 409, 307-318, http://dx.doi.org/10.1016/j.epsl.2014.10.006, 2015.

Herbert, T. D., Lawrence, K. T., Tzanova, A., Peterson, L. C., Caballero-Gill, R., and Kelly, C. S.: Late Miocene global cooling and the rise of modern ecosystems, Nature Geoscience, 9, 843-847,
1095 doi:810.1038/ngeo2813, 2016.

Karas, C., Nurnberg, D., Gupta, A. K., Tiedemann, R., Mohan, K., and Bickert, T.: Mid-Pliocene climate change amplified by a switch in Indonesian subsurface throughflow, Nature Geoscience, 2, 434-438, http://www.nature.com/ngeo/journal/v2/n6/suppinfo/ngeo520_S1.html, 2009.

Karas, C., Nürnberg, D., Tiedemann, R., and Garbe-Schönberg, D.: Pliocene Indonesian Throughflow and
1100 Leeuwin Current dynamics: Implications for Indian Ocean polar heat flux, Paleoceanography, 26, PA2217, 10.1029/2010pa001949, 2011.

Karas, C., Nürnberg, D., Bahr, A., Groeneveld, J., Herrle, J. O., Tiedemann, R., and deMenocal, P. B.: Pliocene oceanic seaways and global climate, Scientific Reports, 7, 39842, 10.1038/srep39842, 2017.

Lawrence, K. T., Liu, Z., and Herbert, T. D.: Evolution of the Eastern Tropical Pacific Through Plio-Pleistocene
1105 Glaciation, Science, 312, 79-83, 2006.

Lawrence, K. T., Herbert, T. D., Brown, C. M., Raymo, M. E., and Haywood, A. M.: High-amplitude variations in North Atlantic sea surface temperature during the early Pliocene warm period, Paleoceanography, 24, 2009.

Lawrence, K. T., Sosdian, S., White, H. E., and Rosenthal, Y.: North Atlantic climate evolution through the Plio-Pleistocene climate transitions, Earth and Planetary Science Letters, 300, 329-342,
1110 https://doi.org/10.1016/j.epsl.2010.10.013, 2010.

Lawrence, K. T., and Woodard, S. C.: Past sea surface temperatures as measured by different proxies—A cautionary tale from the late Pliocene, Paleoceanography, 32, 318-324, 10.1002/2017pa003101, 2017.

Li, L., Li, Q., Tian, J., Wang, P., Wang, H., and Liu, Z.: A 4-Ma record of thermal evolution in the tropical western Pacific and its implications on climate change, Earth and Planetary Science Letters, 309, 10-20, 2011.

1115 Lin, L., Khider, D., Lisiecki, L. E., and Lawrence, C. E.: Probabilistic sequence alignment of stratigraphic records, Paleoceanography, 29, 976–989, 2014.

Liu, J., Tian, J., Liu, Z., Herbert, T. D., Fedorov, A. V., and Lyle, M.: Eastern equatorial Pacific cold tongue evolution since the late Miocene linked to extratropical climate, Science Advances, 5, eaau6060, 10.1126/sciadv.aau6060, 2019.

1120 Locarnini, R. A., A. V. Mishonov, O. K. Baranova, T. P. Boyer, M. M. Zweng, H. E. Garcia, J. R. Reagan, D. Seidov, K. Weathers, C. R. Paver, and I. Smolyar (2018) World Ocean Atlas 2018, Volume 1: Temperature. A. Mishonov Technical Ed.; NOAA Atlas NESDIS 81, 52 pp.

Martinez-Garcia, A., Rosell-Mele, A., McClymont, E. L., Gersonde, R., and Haug, G. H.: Subpolar Link to the Emergence of the Modern Equatorial Pacific Cold Tongue, Science, 328, 1550-1553, 10.1126/science.1184480,
1125 2010.

Martinez-Garcia, A., Rosell-Mele, A., Jaccard, S. L., Geibert, W., Sigman, D. M., and Haug, G. H.: Southern Ocean dust-climate coupling over the past four million years, Nature, 476, 312-315, 2011.

Mashiotta, T. A., Lea, D. W., and Spero, H. J.: Glacial–interglacial changes in Subantarctic sea surface temperature and $\delta$18O-water using foraminiferal Mg, Earth and Planetary Science Letters, 170, 417-432, https://doi.org/10.1016/S0012-821X(99)00116-8, 1999.

McClymont, E. L., Elmore, A. C., Kender, S., Leng, M. J., Greaves, M., and Elderfield, H.: Pliocene-Pleistocene evolution of sea surface and intermediate water temperatures from the Southwest Pacific, Paleoceanography, PA002954, 2016.

Müller, P. J., Kirst, G., Ruhland, G., Storch, I. V., and Rosell-Melé, A.: Calibration of the alkenone paleotemperature index $U^k_{37}{}'$ based on core-tops from the eastern South Atlantic and the global ocean (60°N–60°S), Geochimica et Cosmochimica Acta, 62, 1757–1772, 1998.

Naafs, B. D. A., Stein, R., Hefter, J., Khélifi, N., De Schepper, S., and Haug, G. H.: Late Pliocene changes in the North Atlantic Current, Earth and Planetary Science Letters, 298, 434-442, http://dx.doi.org/10.1016/j.epsl.2010.08.023, 2010.

Nürnberg, D., Müller, A., and Schneider, R. R.: Paleo-sea surface temperature calculations in the equatorial east Atlantic from Mg/Ca ratios in planktic foraminifera: A comparison to sea surface temperature estimates from U37K′, oxygen isotopes, and foraminiferal transfer function, Paleoceanography, 15, 124-134, 10.1029/1999pa000370, 2000.

Petrick, B., McClymont, E. L., Felder, S., Rueda, G., Leng, M. J., and Rosell-Melé, A.: Late Pliocene upwelling in the Southern Benguela region, Palaeogeography, Palaeoclimatology, Palaeoecology, 429, 62-71, http://dx.doi.org/10.1016/j.palaeo.2015.03.042, 2015.

Prahl, F. G., Muehlhausen, L. A., and Zahnle, D. I.: Further evaluation of long-chain alkenones as indicators of paleoceanographic conditions, Geochimica et Cosmochimica Acta, 52, 2303-2310, 1988.

Regenberg, M., Nürnberg, D., Steph, S., Groeneveld, J., Garbe-Schönberg, D., Tiedemann, R., and Dullo, W.-C.: Assessing the effect of dissolution on planktonic foraminiferal Mg/Ca ratios: Evidence from Caribbean core tops, Geochemistry Geophysics Geosystems, 7, Q07P15, doi:10.1029/2005GC001019, 2006.

Rosell-Melé, A., Martínez-Garcia, A., and McClymont, E. L.: Persistent warmth across the Benguela upwelling system during the Pliocene epoch, Earth and Planetary Science Letters, 386, 10-20, http://dx.doi.org/10.1016/j.epsl.2013.10.041, 2014.

Sánchez-Montes, M. L., McClymont, E. L., Lloyd, J. M., Müller, J., Cowan, E. A., and Zorzi, C.: Late Pliocene Cordilleran Ice Sheet development with warm Northeast Pacific sea surface temperatures, Clim. Past Discuss. (accepted), 2019, 1-23, 10.5194/cp-2019-29, 2019.

Seki, O., Foster, G. L., Schmidt, D. N., Mackensen, A., Kawamura, K., and Pancost, R. D.: Alkenone and boron-based Pliocene pCO2 records, Earth and Planetary Science Letters, 292, 201-211, 2010.

Tian, J., Pak, D. K., Wang, P., Lea, D., Cheng, X., and Zhao, Q.: Late Pliocene monsoon linkage in the tropical South China Sea, Earth and Planetary Science Letters, 252, 72-81, https://doi.org/10.1016/j.epsl.2006.09.028, 2006.

Tierney, J. E., and Tingley, M. P.: BAYSPLINE: A New Calibration for the Alkenone Paleothermometer, 33, 281-301, 10.1002/2017pa003201, 2018.

Tierney, J. E., Malevich, S. B., Gray, W., Vetter, L., and Thirumalai, K.: Bayesian calibration of the Mg/Ca paleothermometer in planktic foraminifera, Paleoceanography and Paleoclimatology, 31, https://doi.org/10.1029/2019PA003744, 10.1029/2019pa003744, 2019.

Tzanova, A., and Herbert, T. D.: Regional and global significance of Pliocene sea surface temperatures from the Gulf of Cadiz (Site U1387) and the Mediterranean, Global and Planetary Change, 133, 371-377, https://doi.org/10.1016/j.gloplacha.2015.07.001, 2015.

Wara, M. W., Ravelo, A. C., and Delaney, M. L.: Permanent El Nino-Like Conditions During the Pliocene Warm Period, Science, 309, 758-761, 2005.